

# RiceChain-Plus: an enhanced framework for blockchain-based rice supply chain systems-ensuring security, privacy, and efficiency

Bello Musa Yakubu[1], Abdullah Abdulrahman Alabdulatif[2] and Pattarasinee Bhattarakosol[1]

[1] Department of Mathematics and Computer Science, Faculty of Science, Chulalongkorn University, Bangkok, Thailand
[2] Department of Cybersecurity, College of Computer, Qassim University, Buraydah, Saudi Arabia

## ABSTRACT

The rice supply chain is a complex system that demands effective management to ensure reliability and efficiency, given the involvement of multiple stakeholders. Blockchain technology, with its decentralized and tamper-resistant nature, offers a promising solution for improving transparency, traceability, and credibility in agricultural supply chains. However, existing blockchain systems face several technological challenges, including security vulnerabilities, privacy concerns, and performance limitations. To address these issues, this article presents RiceChain-Plus, an enhanced architecture that incorporates a private Ethereum blockchain, proof of authority (PoA) consensus mechanism, mutual authentication, zero-knowledge proofs (ZKPs), a hybrid role-based access control (RBAC) and attribute-based access control (ABAC) system, and one-way hash functions. This approach enhances the rice supply chain's security, privacy, and efficiency by safeguarding sensitive data and ensuring confidentiality. Performance assessments show that RiceChain-Plus surpasses existing benchmark models, achieving the lowest average execution costs (44,634 gas), reduced energy consumption (9.38828E−05 J), higher throughput (0.071201 transactions/s), faster execution (44.5 ms), and quicker transaction times (14.045 s), while also improving scalability. A comprehensive security analysis further confirms the framework's resilience against various cyberattacks. These results highlight RiceChain-Plus as a secure, efficient, and effective solution for optimizing rice supply chain operations.

## INTRODUCTION

The rice supply chain is a complex and interdependent system involving several stakeholders, such as growers, processors, distributors, and retailers. Ensuring the integrity, security, and efficiency of the supply chain is important for safeguarding food safety, reducing waste, and strengthening consumer trust (*Md Nasir et al., 2024*; *Arfin et al., 2024*). Blockchain technology, defined by its decentralised structure,

Corresponding authors
Abdullah Abdulrahman Alabdulatif, a.alabdulatif@qu.edu.sa
Pattarasinee Bhattarakosol, pattarasinee.b@chula.ac.th

transparent operations, and immutable data, offers possible solutions for enhancing the traceability and dependability of agricultural supply chains (*Li et al., 2024*; *Md Nasir et al., 2024*). The use of blockchain technology in rice supply chain management has significant technical problems related to security, privacy, and efficiency (*Md Nasir et al., 2024*; *Rahman, Ooi & Lee, 2025*; *Surucu-Balci, Iris & Balci, 2024*).

Scholars have shown that blockchain technology can be combined with other several forms of security and privacy enhancement techniques to improve traceability, safety, privacy, and efficiency in blockchain-based agricultural supply chains (*Faisal et al., 2024*; *Surucu-Balci, Iris & Balci, 2024*). Similarly, studies demonstrate that using several blockchain and consensus mechanisms is beneficial for product monitoring, participant identity management, and system performance optimisation (*Altaf et al., 2022*). However, challenges such as increased energy consumption and vulnerability to substantial threats, including replay, side-channel, identity, privacy-related threats, and other forms of denial-of-service attacks, continue to exist.

The integration of blockchain technology with machine learning and cryptography methods can significantly enhance the validity, dependability, and security of data and user privacy. *Zhang et al. (2023)* suggested a system that integrates blockchain technology with machine learning to tackle anomalous data management. *Peng et al. (2022b)* developed a cross-chain model that combines blockchain technology with unsupervised machine learning to enhance data exchange and operational efficiency within the rice supply chain. Nonetheless, issues persist, such as delay and energy consumption (*Arfin et al., 2024*; *Zhang, Jia & Chen, 2024*).

In a nutshell, blockchain technology has been increasingly adopted in agricultural supply chains for its ability to ensure transparency, security, and efficiency. However, existing solutions face critical challenges, including scalability limitations, energy consumption, and efficiency issues, along with security and privacy concerns. The primary objective of this work is to present RiceChain-Plus, a blockchain architecture aimed at addressing the above critical challenges in the rice supply chain. The system ensures secure and efficient management of rice supply chain activities by integrating a private blockchain architecture with advanced cryptographic techniques, such as zero-knowledge proofs (ZKP). The study further seeks to enhance stakeholder trust and supply chain transparency using immutable data and dynamic access control mechanisms. The study focuses on protecting sensitive data along the rice supply chain, including activities such as planting, harvesting, processing, storage, and transportation. RiceChain-Plus clearly addresses issues such as data integrity, unauthorised access, and other security, privacy, and efficiency constraints that emerge throughout these phases.

RiceChain-Plus uses a resource-efficient proof of authority (PoA) consensus and ZKPs to improve scalability, data security, and privacy. The approach is different from other blockchain applications in the food and agricultural sector that rely on consensus mechanisms like proof of work or limited privacy-preserving techniques. This combination especially targets the challenges within the rice supply chain, including unlawful access and manipulation. The study uses a private Ethereum blockchain with one-way hash functions and role-based access control (RBAC) and attribute-based access

control (ABAC) to make rice supply chains safer, more private, and more efficient. This ensures food safety and security. By implementing these advanced security measures, the study aims to provide a robust framework that protects sensitive data and fosters transparency among stakeholders. Ultimately, this approach seeks to improve traceability in the rice supply chain, ensuring that consumers can trust the integrity of the products they purchase.

The following contributions are made in the article:

- Use of a hybrid RBAC and ABAC and one-way hash functions enhances security against cyberattacks and unauthorized access.
- RiceChain-Plus ensures privacy of sensitive data using robust cryptographic methods such as zero-knowledge proofs. PoA consensus method and private Ethereum blockchain minimize computational burden, enabling faster, energy-efficient transaction processing.
- Comprehensive framework integrates multiple Internet of Things (IoT) devices and sensors, providing holistic monitoring and product quality preservation.
- Architecture enables easy expansion and customization, supporting growing stakeholders and transactions.
- Burrows–Abadi–Needham (BAN) logic was used to perform the proof of correctness for mutual authentication and key exchange phase.

The subsequent sections of this article are organized as follows. "Literature Review" comprises of a comprehensive overview of the current literature. "Network Model" provides a discussion on the network model including discussion on sensory nodes in the rice supply chain network, model, and security and privacy requirements. "Proposed Security and Privacy Fortifications in RiceChain-Plus Framework" encompasses the step-by-step discussions on the proposed model. "Security Analysis" provides a comprehensive security analysis on the proposed model. "Performance Evaluation and Discussions" presents the experimental findings and assessment. The article concludes with an overview of the findings and outlines future objectives in section seven.

## LITERATURE REVIEW

The use of blockchain technology into agricultural supply chains has attracted considerable interest for improving traceability, security, and efficiency (*Chang, Iakovou & Shi, 2020*). Nonetheless, other technical obstacles remain, including privacy, scalability, energy consumption, and security concerns. This section classifies previous research according to these issues and discusses how RiceChain-Plus resolves them.

### Privacy and security

Maintaining data privacy and security is a significant problem in blockchain-based agricultural systems. Several studies use cryptographic methods to improve data security:

- *Multi-Party Encryption: Xu et al. (2022)* introduced a grain and oil food traceability system that integrates blockchain technology with multi-party hybrid encryption to

ensure data security. The system, however functional, encountered scalability challenges and reliance on human data entry (*Duan, Pang & Lin, 2023*; *Md Nasir et al., 2024*).

- *Blockchain and Asymmetric Cryptography: Shivendra et al. (2021)* examined the use of blockchain and asymmetric cryptography to improve traceability in food and agricultural supply chains. Notwithstanding progress, challenges regarding data integrity and verification continued (*Arfin et al., 2024*).
- *Consortium Blockchain Models: Saranya & Maheswari (2023)* introduced a PoTx-based consortium blockchain model for agricultural traceability. Notwithstanding enhanced efficiency, vulnerabilities like DDoS attacks and energy inefficiencies remained (*Yakubu et al., 2022*).

   *RiceChain-Plus Contribution*: RiceChain-Plus ensures safe data validation and keeps private data safe by combining Zero-Knowledge Proofs (ZKP) and cryptographic hashing (SHA-256). These procedures address issues pertaining to privacy violations and enhance overall system robustness.

## Scalability

The scalability constraints of blockchain often impede its use in high-volume agricultural supply networks.

- *Consensus Mechanisms: Yakubu et al. (2022)* and *Khanna et al. (2022)* established a rice supply chain system using Ethereum's Proof of Authority (PoA) consensus, enhancing scalability but encountering issues pertaining to decentralisation (*Farooq et al., 2024*).
- *Cross-Chain Solutions: Peng et al. (2022b)* developed a Multi-Blockchain Rice Refined Supervision Model (MBRRSM) that incorporates multi-party encryption and the Supervision Proof of Peers (SPOP) consensus mechanism. Nevertheless, it demonstrated lowered scalability and throughput (*Han & Fang, 2024*; *Md Nasir et al., 2024*).

   *RiceChain-Plus Contribution*: Utilising a lightweight proof-of-authority consensus method and fog nodes for gathering data, RiceChain-Plus addresses scaling challenges by minimising computational overhead and facilitating quick transaction validation.

## Energy consumption

Excessive energy usage continues to be an important roadblock to the deployment of blockchain technology in agricultural supply chains.

- *Integration of IoT and Blockchain: Saranya & Maheswari (2023)* used IoT devices with a hybrid blockchain architecture using Proof of Transaction (PoTx) consensus to improve efficiency. Nonetheless, energy consumption and vulnerability to DDoS attacks were significant drawbacks (*Altaf et al., 2022*; *Yakubu et al., 2023*).
- *Hydroponic Vegetable Supply Chains: Suroso, Rifai & Hasanah (2021)* developed a blockchain-based system for hydroponic vegetable supply chains that integrates IoT devices. Although effective in improving traceability, it encountered increased energy consumption (*Rahman, Ooi & Lee, 2025*).

*RiceChain-Plus Contribution:* RiceChain-Plus reduces energy consumption by implementing fog nodes and optimising cryptographic processes, making it a sustainable option for agricultural supply chains.

## Data integrity and storage optimisation

Ensuring data integrity and optimising storage are essential for sustaining reliable and efficient systems.

- *IPFS Integration: Wang et al. (2021)* and *Yao & Zhang (2022)* used the InterPlanetary File System (IPFS) to mitigate storage challenges and enhance data retrieval efficiency. Despite their effectiveness, these systems encountered substantial installation expenses and susceptibility to collisions (*Zhang, Jia & Chen, 2024*).
- *Merkle Trees:* Cryptographic constructs such as Merkle Trees were used to ensure data privacy and integrity. Nonetheless, difficulties such as collision attacks continued to exist (*Bosona & Gebresenbet, 2023*).

*RiceChain-Plus Contribution:* Using SHA-256 cryptographic hashing and the assurance of data immutability, RiceChain-Plus upholds data integrity. The private blockchain design minimises overhead storage and improves data security.

## Advanced integration with machine learning

The integration of machine learning with blockchain presents innovative methods for anomaly detection and data integrity (*Cherbal et al., 2024*; *Li et al., 2022*).

- *Three-Tier Systems: Zhang et al. (2023)* integrated blockchain technology with machine learning models, such as isolation forests and random forests, to improve data dependability. The system, albeit successful, had challenges generalising to new data and was susceptible to false data injection attacks (*Salama & Eassa, 2024*).
- *Cross-Chain Data Management: Peng et al. (2022a)* proposed a cross-chain methodology using unsupervised machine learning approaches to enhance data interchange. Notwithstanding progress, latency and energy consumption continued to pose issues (*Li et al., 2024*).

RiceChain-Plus Contribution: RiceChain-Plus mainly focusses on blockchain and cryptography methods. However, in the future, improvements could include machine learning models to make finding anomalies and validating data easier, which would help fix problems with current systems.

In conclusion, despite notable progress in agricultural traceability with blockchain in previous research, concerns like privacy, scalability, energy consumption, and data integrity remain unresolved. These problems are fixed in RiceChain-Plus by using fog nodes, cryptographic hashing, zero-knowledge proofs, and a streamlined proof of authority consensus process. These advances render it a resilient solution tailored to the specific requirements of the rice supply chain. Table 1 provides a summary of the relevant studies, classified by the approach used, the objectives sought, and the limitations faced. To

**Table 1 Summary of the related works.**

| Reference | Technique used | Objectives | Limitations |
|---|---|---|---|
| *Saranya & Maheswari (2023)* | Consortium blockchain, Proof of Transaction (PoTx) | Improve product tracking, maintain participant identities, enhance system efficiency | Energy consumption, vulnerability to DDoS attacks |
| *Peng et al. (2022b)* | Blockchain, Supervision Proof of Peers (SPOP) | Enhance traceability, improve security | Scalability, operational speed, energy consumption, vulnerability to attacks |
| *Suroso, Rifai & Hasanah (2021)* | Blockchain | Enhance supply chain security and transparency | Fraudulent data injection, high energy consumption |
| *Yang et al. (2021)* | Blockchain, key-value database | Address issues in rice supply chain management, improve transparency | Restricted decentralization, insufficient deep analysis |
| *Zhang et al. (2023)* | Blockchain, machine learning | Improve data authenticity and reliability | Generalization to new data, vulnerability to fake data injection attacks |
| *Xu et al. (2022)* | Blockchain, Unsupervised Machine Learning | Improve data sharing and technological efficiency | Latency, energy consumption, vulnerability to attacks |
| *Khanna et al. (2022)* | Blockchain, IoT integration | Enhance safety and quality of dairy products, optimize supply chain processes | Lack of in-depth analysis of experimental outcomes, resilience against network threats |
| *Peng et al. (2022a)* | Blockchain, Multi-Party Encryption | Enhance quality and safety of rice | Lower throughput, high resource consumption |
| *Shivendra et al. (2021)* | Blockchain, asymmetric cryptography | Improve traceability and safety in food and agricultural supply chains | Data integrity and authentication concerns |
| *Yao & Zhang (2022)* | Blockchain, InterPlanetary File System (IPFS) | Enhance Agricultural Traceability System for food production | High implementation costs, susceptibility to collision attacks |
| *Yakubu et al. (2022)* | Ethereum blockchain, PoA consensus method | Improve traceability and monitoring in the rice supply chain | Location monitoring attacks, limited decentralization |
| *Patel et al. (2024)* | Blockchain, secure multi-party algorithms | Improve traceability and security in agricultural supply networks | Scalability, implementation cost |
| *Wang et al. (2021)* | Blockchain, InterPlanetary File System (IPFS) | To address problems in agrifood supply chain management, improve transparency | Limited decentralization, high implementation costs, susceptibility to attacks |

implement a practical blockchain solution addressing the challenges highlighted in the literature, we developed a network model tailored to the rice supply chain, as discussed in the following section.

# NETWORK MODEL

The rice supply chain network operates on a private Ethereum blockchain, linking various stakeholders such as seed suppliers, farmland operators, grain elevators, processors, distributors, retailers, and consumers, as seen in Fig. 1. Each stakeholder group is represented by a cluster head (CH) that oversees a subnetwork of sensory devices. These sensory devices, with constrained computational capabilities, observe and relay real-time parameters like temperature, humidity, and product quality. The CHs, as higher-capacity devices, consolidate this data, do preliminary validation, and transmit the processed information to the blockchain network.

## Fog nodes serving as intermediaries

Fog nodes in RiceChain-Plus, denoted as CHs, serve as intermediaries between sensor devices and the blockchain network, significantly improving efficiency. Besides

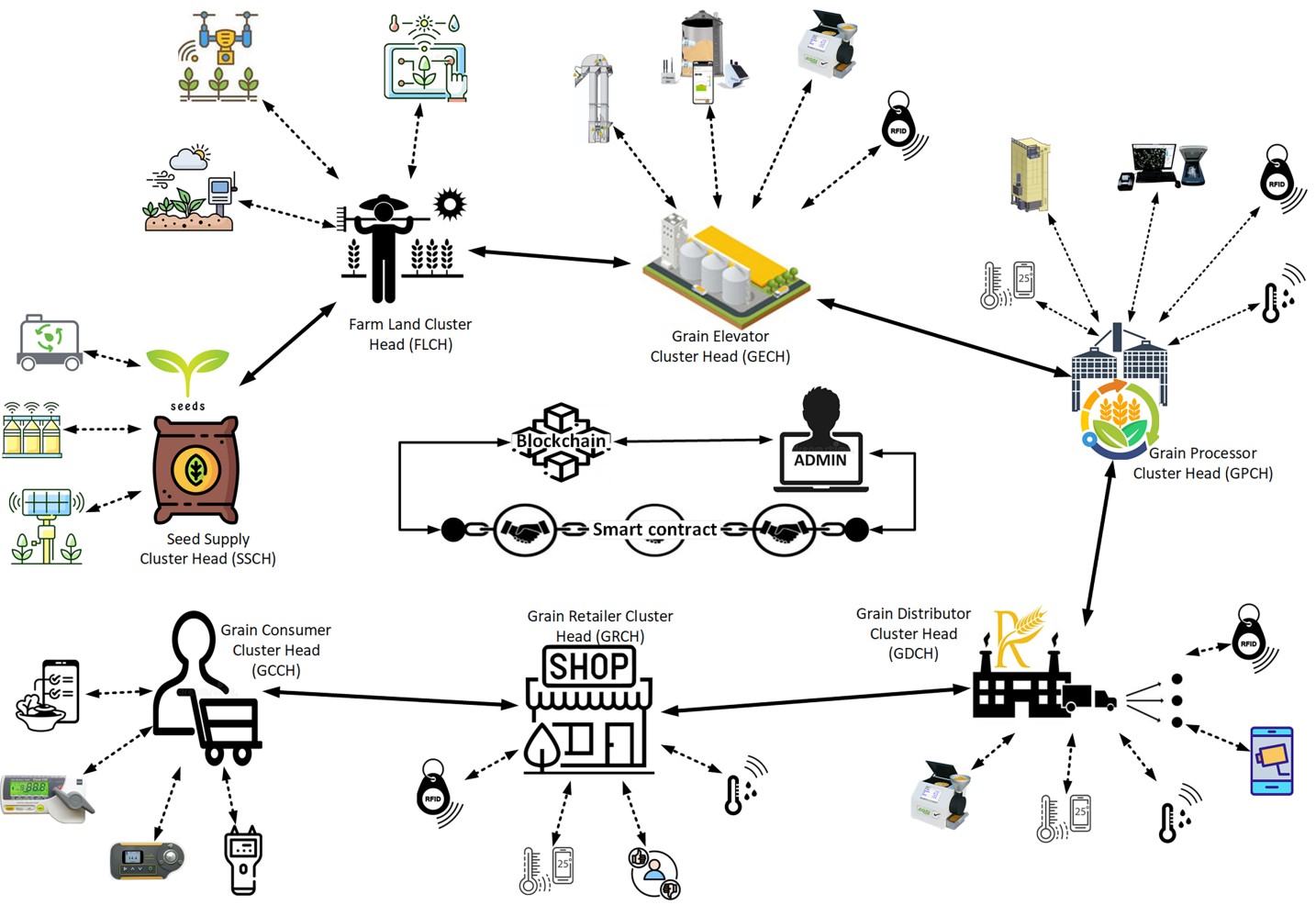

**Figure 1 Proposed secured RiceChain-Plus framework.** The RiceChain-Plus framework's high-level architecture illustrates the interaction flow among stakeholders, devices, and blockchain components. The "Network Model" section provides details on the cryptographic primitives (*e.g.*, SHA-256 hashing, zero-knowledge proofs, mutual authentication) used at various stages.

aggregating data, fog nodes provide essential cryptographic functions to guarantee secure interaction and data integrity:

- SHA-256 Hashing: Sensor device data is hashed using the SHA-256 cryptographic method prior to transmission to the blockchain. This guarantees data integrity by allowing stakeholders to confirm that the data has remained unaltered throughout transmission or storage.

- Zero-Knowledge Proofs (ZKP): Fog nodes use ZKP to verify data integrity while preserving confidential information. The sensory data on product quality is validated for accuracy while maintaining confidentiality, hence assuring privacy and security across the network.

- Mutual Authentication: Mechanisms for mutual authentication are used between the devices in the system to avert impersonation and unauthorised access. Nonces and

timestamps are used in the authentication process to prevent replay attacks and maintain session freshness.

These cryptographic procedures, together with data compression techniques, reduce transmission costs, improve security, and ensure secure transfer to the blockchain. The decentralised architecture of fog nodes decreases latency and prevents bottlenecks, particularly in resource-constrained settings.

## Stakeholder-specific communication

Each CH, representing a stakeholder, is allocated a distinct Ethereum account (EA) for secure communication within the network. Transactions are digitally verified by the CHs using their private keys, hence assuring authenticity and non-repudiation. The central node, known as the admin or gateway node, functions as the principal aggregator, authenticator, and mining node. The admin node does the following functions:

- Transaction Management: The admin verifies transactions submitted by CHs, ensuring that only approved parties can interact with the blockchain.
- Block Addition: The admin node incorporates confirmed transactions into the blockchain ledger *via* the Proof of Authority (PoA) consensus process.
- Key Management: The admin allocates shared cryptographic keys to CHs at the initial registration stage to enable secure communication.

## Smart contracts and blockchain operations

Smart contracts are integrated into the blockchain to automate transaction processing and allow stakeholders to monitor and track developments using mobile apps. These contracts implement stakeholder-specific regulations, including access control policies grounded on Role-Based Access Control (RBAC) and Attribute-Based Access Control (ABAC), to dynamically manage permissions.

Data saved on the blockchain is session-specific, indicating its removal post-product expiry to alleviate computational burden and facilitate further on-chain transactions. Data is encrypted and structured according to stakeholder-specific regulations to preserve privacy and reduce redundancy. Only authorised individuals are permitted to decipher and access pertinent information, hence preserving the secrecy of sensitive documents.

## Sensory nodes in the rice supply chain network

Sensory nodes are essential in maintaining the quality and efficiency of the value chain, from seed cultivation to consumption (*Firdaus, Rahmadika & Rhee, 2021*; *Nguyen & Mohammadi, 2023*; *Vijayan et al., 2021*). These nodes, deployed at various stages, collect critical data to optimize processes and ensure the quality of the product. They gather information at various stages of the process, such as monitoring environmental factors, seed status, and stock control in the seed supply domain. Sensory nodes in the farmland stage enhance growth characteristics and productivity by transmitting data to the FLCH. Sensory nodes in grain elevators preserve and maintain rice quality by transmitting data to the GECH for analysis. Sensory nodes in grain processing plants regulate quality,

productivity, and safety, with the GPCH receiving the information. Sensory nodes in grain distribution domain monitor storage and transportation conditions, transmitting data to the GDCH. Sensory nodes in the retail domain evaluate the safety and quality of rice for consumers by transmitting information to the GRCH. Sensory nodes in the consumer domain ensure the safety and quality of rice, using sensors for temperature, humidity, gas, light, weight, vibration, pests, microbial, and optical detection. Strategically deploying these sensory nodes across the entire rice supply chain will maintain quality, safety, and efficiency at every level.

### Adversary model

In the context of the RiceChain-Plus framework, the adversary model comprises various potential threats that aim to compromise the security, privacy, and efficiency of the blockchain-based rice supply chain system. The key types of adversaries considered in this model include:

*i. External Adversaries:*

a. Eavesdroppers: These adversaries aim to intercept communication between the stakeholders in the supply chain. They might try to capture data such as sensor readings, transaction details, or personal information.

b. Attackers on Network Nodes: These adversaries attempt to compromise network nodes such as the cluster heads (CHs) or the gateway ($G$). They might aim to manipulate the data, perform denial-of-service (DoS) attacks, or gain unauthorized access to sensitive information.

*ii. Internal Adversaries:*

a. Malicious Insiders: These adversaries include compromised or malicious stakeholders within the supply chain, such as a dishonest farmer or processor. They might aim to tamper with the data, forge transactions, or disrupt the supply chain processes.

b. Compromised Devices: Sensory devices or cluster heads that are compromised could send false data, leading to incorrect information being recorded in the blockchain.

### Security and privacy requirements

i. *Confidentiality:* Encrypts sensitive data during transmission and storage.
ii. *Integrity:* Uses robust hashing algorithms for data verification.
iii. *Authentication:* Employs mutual authentication mechanisms for accessing sensitive data.
iv. *Authorization:* Defines clear access control policies restricting stakeholder permissions.
v. *Non-repudiation:* Records transactions in an immutable ledger.

    *vi. Replay Attack Prevention:* Uses timestamp-based checks and nonce mechanisms to prevent replay attacks.

   *vii. Scalability:* Design the system to handle growing transactions and stakeholders efficiently.

  *viii. Privacy:* Implements privacy-preserving techniques for sensitive information verification.

    *ix.* Anonymizes stakeholders' private data and shares only necessary information among supply chain participants.

## PROPOSED SECURITY AND PRIVACY FORTIFICATIONS IN RICECHAIN-PLUS FRAMEWORK

The proposed framework is designed with two levels of security. The first level of security consists of four phases: registration, user device login, mutual authentication, and user password replacement protocols. While the second level consists of detailed process on how the framework achieves, authorization and privacy. Table 2 provide the overview of adversary types, and their countermeasures used in this study; while Table 3 provides the descriptions of the symbols used in this model.

### First security level: initialization and mutual authentication of stakeholders and their associated devices

This level provides detailed processes involved in the protocols used to validate stakeholders and their associated devices in the proposed framework.

#### Registration phase

Stage 0: The admin $\mathcal{G}$ register all stakeholders' cluster heads $CH = \{CH_1, CH_2, CH_3, \ldots CH_i\}$ and the corresponding mobile device Identifications $D = \{d_1, d_2, d_3, \ldots d_i\}$. Each CH and the admin $\mathcal{G}$ are identified by their unique Ethereum address (EA).

Stage 1: Each CH generates a private key and a corresponding public key using elliptic curve cryptography parameters. Using its private key and the admin's public key, each CH computes it secretes key $CH_{skey}$ using Elliptic Curve Diffie-Hellman (ECDH) algorithm.

    Using this secrete key, the admin's Ethereum address $\mathcal{G}_{EA}$, and its Ethereum address $CH_{EA}$, the CH generates, and stores shared secrete keys with $\mathcal{G}$ as follow:

$$\mathcal{G}_{skey} = h\big(\mathcal{G}_{EA}\|CH_{skey}\big) \tag{1}$$

$$CH_{share} = h\big(CH_{EA}\|CH_{skey}\big) \tag{2}$$

The computed keys are securely stored in a Hardware Security Module (HSM) to prevent unauthorized access.

Stage 2: Each CH $CH_i$ then chooses an identification $S_{id}$ to each sensor node $\mathcal{S}n_i$ in its cluster and store it.

Stage 3: Each CH shares $\mathcal{G}_{skey}$, $CH_{share}$, and $S_{id}$ with $\mathcal{G}$.

**Table 2 Adversary countermeasures.**

| Adversary type | Threats | Affected process stage | Countermeasures |
|---|---|---|---|
| Malicious insiders | Data tampering, transaction forgery, disruption of supply chain processes | Data aggregation and blockchain transaction validation | Cryptographic hashing, mutual authentication, role-based and attribute-based authorization (RBAC/ABAC) |
| External eavesdroppers | Interception of sensitive communication during data transmission | Data transmission | Encryption (SHA-256), timestamps, session-specific nonces |
| Replay attackers | Resending legitimate but expired messages to gain unauthorized access or disrupt processes | Authentication phase and data validation | Timestamps, session-specific nonces, freshness checks |
| Attackers on network nodes | Data manipulation, denial-of-service (DoS) attacks, unauthorized access | Node-to-node communication and validation stages | Timestamps, session-specific nonces, authorization processes (RBAC/ABAC), data validation (Zero-Knowledge Proofs) |
| Compromised devices | Injecting false or tampered data into the supply chain | Data collection and aggregation | Data validation (Zero-Knowledge Proofs), secure mutual authentication |

**Table 3 Table of symbols.**

| Symbol | Description |
|---|---|
| $\mathcal{G}$ | Admin or gateway device managing the blockchain network. |
| $\mathcal{G}_{EA}$ | Admin's Ethereum address. |
| $CH$ | Cluster head responsible for aggregating and validating data. |
| $CH_{EA}$ | Ethereum address of $CH$. |
| $CH_{skey}$ | Secrete key of $CH$. |
| $d_i$ | Unique identifier for a mobile device in the network. |
| $S_{id}$ | Unique identifier for a sensor node in the network. |
| $Sn_i$ | A given sensor node in a network. |
| $E_{psw}$ | Encapsulated hashed password for secure user authentication. |
| $SH_{id}$ | Stakeholder unique identification. |
| $SH_{psw}$ | Stakeholder strong password. |
| $E_{id}$ | Encapsulated identification. |
| $\varphi_1, \varphi_2, \varphi_3$ | Parameters derived during the password hashing and authentication process. |
| $r_1, r_2, r_3, r_4, r_5$ | Random numbers (nonces) generated to ensure freshness and prevent replay attacks. |
| $T_1, T_2, T_3, T_4, T_5, T_6$ | Timestamps used to validate the timing of transactions and authentication. |
| $\beta_1, \beta_2, \beta_3$ | Session-specific identifiers used in the authentication process. |
| $\gamma_1, \gamma_2, \gamma_3$ | Authentication parameters exchanged between $\mathcal{G}$ and $CH$. |
| $\rho_1, \rho_2$ | Parameters generated by $CH$ for message validation. |
| $\sigma_1, \sigma_2$ | Parameters computed by $\mathcal{G}$ for final authentication and verification. |
| $\mathcal{G}_{skey}$ | Shared secret key between $\mathcal{G}$ and $CH$. |
| $CH_{share}$ | Shared key derived by the cluster head. |
| $h(X)$ | Cryptographic hash function, typically SHA-256, applied to input X. |
| $II$ | Concatenation operator used in cryptographic computations. |
| $\oplus$ | XOR operator used for encoding and secure computation. |
| $\#(X)$ | Indicates freshness of X (*i.e.*, X has not been used before). |
| $\theta$ | Maximum allowable transmission delay to prevent replay attacks. |

Given that the sensing data provided by all sensor nodes in the model are only for consumption of the registered users (stakeholders) in the supply chain network, each stakeholder will need to register its mobile device with which it will access these data when released. The process is outlined in the following stages.

Stage 4: The stakeholder selects a unique identification $SH_{id}$ and a strong password $SH_{psw}$ for their mobile device. The device generates a random number $r_1$, selects its corresponding mobile device ID $d_i \in D$ and computes the encapsulated password hash:

$$E_{psw} = h\left(r_1 \oplus SH_{psw} \oplus d_i\right) \tag{3}$$

Then, $E_{psw}$ and $SH_{id}$ are shared with the $\mathcal{G}$.

Stage 5: Upon receiving $E_{psw}$ and $SH_{id}$, $\mathcal{G}$ generate another random number $r_2$ and then compute the following values at the timestamp $T_1$:

$$\varphi_1 = h\left(E_{psw} \| T_1\right) \tag{4}$$

$$\varphi_2 = h\left(E_{psw} \| \mathcal{G}_{EA}\right) \tag{5}$$

$$\varphi_3 = h(\varphi_1 \| r_2 \| \mathcal{G}_{EA}) \oplus h\left(E_{psw} \| T_1\right) \tag{6}$$

$\mathcal{G}$ stores these values ($\varphi_1$, $\varphi_2$, $\varphi_3$, $r_2$, $T_1$) in its memory and then share them with the mobile device.

Stage 6: The mobile device stores these values and computes its encapsulated identification ($E_{id}$) as follow:

$$E_{id} = h\left(SH_{psw} \| SH_{id}\right) \oplus r_1 \tag{7}$$

The mobile device then stores the value ($E_{id}$) in its memory. The summary of the registration phase is provided in Fig. 2.

### User device login phase

Stage 1: The stakeholder enters $SH_{id}$ and $SH_{psw}$ into the mobile device with.

Stage 2: The user then selects the cluster head Ethereum address ($CH_{EA}$) of the cluster from which the requesting data is and then enters the address of the admin $\mathcal{G}_{EA}$.

Stage 3: With reception of these information, the mobile device then computes the following values:

$$r_1^* = E_{id} \oplus h\left(SH_{psw} \| SH_{id}\right) \tag{8}$$

$$E_{psw}^* = h\left(r_1^* \oplus SH_{psw} \oplus d_i\right) \tag{9}$$

$$\varphi_2^* = h\left(E_{psw}^* \| \mathcal{G}_{EA}\right) \tag{10}$$

Stage 4: The device then checks if $\varphi_2^*$ and $\varphi_2$ are equal. If the result is true, then the $SH_{id}$ and $SH_{psw}$ of the stakeholder user are verified and then it moves to the next stage. Else, the session is terminated.

Stage 5: In this stage, the mobile device generates a random number $r_3$ and then compute the following values at timestamp $T_2$.

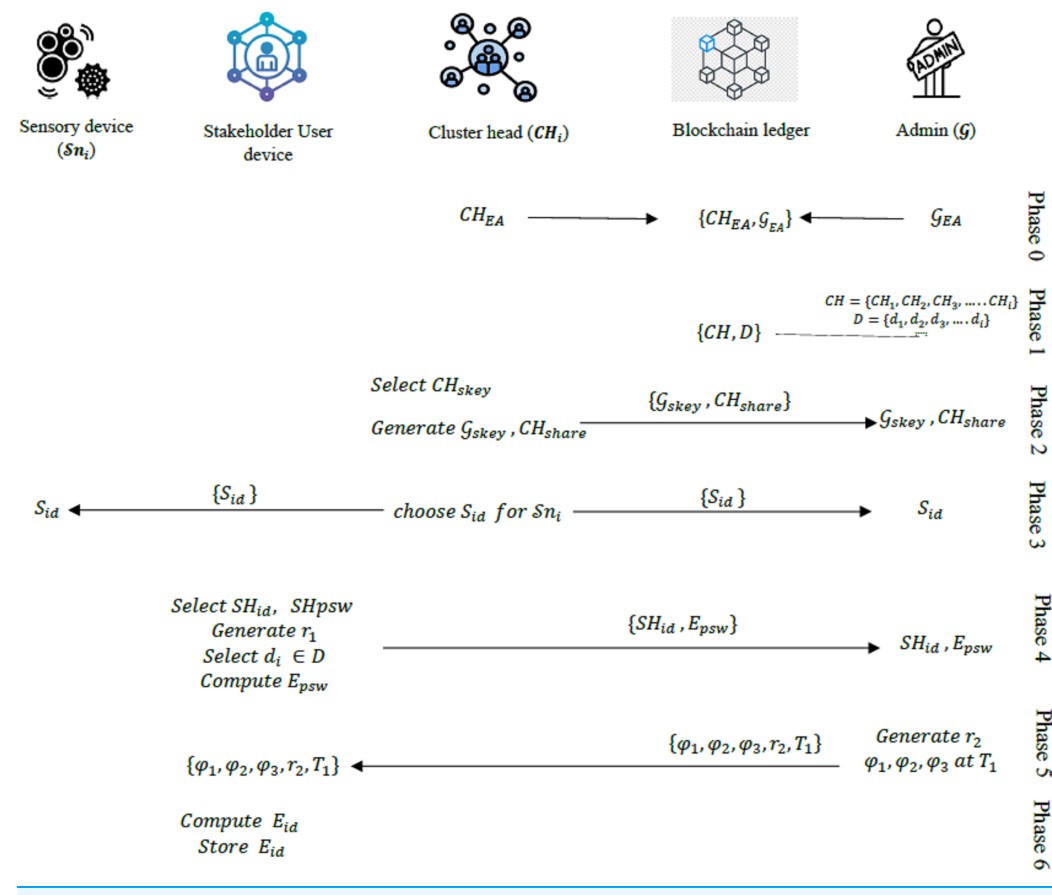

**Figure 2 Overview of the registration phase.**

$$\beta_1 = \varphi_3 \oplus h(E_{psw} \| T_1) \tag{11}$$
$$\beta_2 = h(T_2 \| r_3 \| \beta_1 \| \mathcal{G}_{EA}) \tag{12}$$
$$\beta_3 = (r_3 \| T_2) \oplus \beta_1 \tag{13}$$

The mobile device then submits the values $(CH_{EA}, \beta_2, \beta_3)$ to the $\mathcal{G}$. The summary of the user device login phase is provided in Fig. 3.

### Mutual authentication phase

The mutual authentication as a key important protocol in this work, was selected for its capacity to thwart impersonation and replay attacks by validating the identities of both parties *via* the use of nonces and timestamps. The following are the stages involved in this work.

#### 1. Steps Executed by the Admin Gateway ($\mathcal{G}$)

i. $\mathcal{G}$ receives $\beta_2, \beta_3$ and computes the following at timestamp $T_3$:

$$\beta_1^* = \varphi_3 \oplus h(E_{psw} \| T_1) \tag{14}$$
$$\beta_1^* = \beta_3 \oplus (r_3 \| T_2) \tag{15}$$

ii. $\mathcal{G}$ checks if $T_3 - T_2 < \theta$. If true, the session continues; otherwise, it terminates.

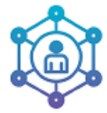
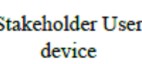

Stakeholder User
device

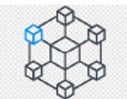
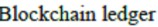

Blockchain ledger

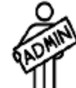

Admin ($\mathcal{G}$)

Input $SH_{id}$ , $SH_{pswe}$

Seclect $CH_i$ , $\mathcal{G}$ address

Compute $r_1^*, E_{psw}^*, \varphi_2^*$
Verify $\varphi_2^*$ with $\varphi_2$

Generate $r_3 , \beta_1, \beta_2, \beta_3$ at $T_2$

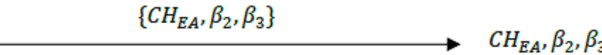

$\{CH_{EA}, \beta_2, \beta_3\}$

$CH_{EA}, \beta_2, \beta_3$

Phase 1  Phase 2  Phase 3  Phase 4  Phase 5

**Figure 3 Overview of the user device login phase.**

iii. $\mathcal{G}$ computes:

$$\beta_2^* = h\left(r_3\|T_2\|\beta_1^*\|\mathcal{G}_{EA}\right) \tag{16}$$

iv. If $\beta_2^* = \beta_2$, the mobile device is verified. Otherwise, the session terminates.

v. $\mathcal{G}$ generates a random number $r_4$ and computes:

$$\gamma_1 = h(CH_{EA}\|\varphi_1\|CH_{share}\|r_4\|T_3) \tag{17}$$

$$\gamma_2 = (r_3\|T_3\|r_4) \oplus CH_{share} \tag{18}$$

$$\gamma_3 = \varphi_1 \oplus h(CH_{EA}\|h(r_4)r_3) \tag{19}$$

vi. $\mathcal{G}$ stores and transmits $\gamma_1, \gamma_2, \gamma_3$ to the cluster head (CH).

vii. Upon receiving $\omega, \rho_1, \rho_2$ from CH, $\mathcal{G}$ performs the following at timestamp $T_5$:

Computes$(\|r_5\|T_4) = \rho_2 \oplus r_4 \tag{20}$

Verifies if $T_5 - T_4 < \theta$. If false, the session terminates.

Computes $\rho_1^* = h(T_4\|r_5\|CH_{EA}\|CH_{share}\|T_3\|\omega), \tag{21}$

If $\rho_1 = \rho_1^*$, CH is verified. Otherwise, the session terminates.

viii. $\mathcal{G}$ computes:

$$\sigma_1 = h(\omega\|\varphi_1\|r_4\|T_5\|\gamma_1) \tag{22}$$

$$\sigma_2 = h(r_5\|r_4\|T_5) \oplus r_3 \tag{23}$$

ix. $\mathcal{G}$ shares $\gamma_1, \sigma_1$, and $\sigma_2$ with the user device.

### 2. Steps Executed by the Cluster Head (CH)

i. CH receives $\gamma_1, \gamma_2, \gamma_3$ and computes the following at timestamp $T_4$:

$$(r_3\|r_4\|T_3) = \gamma_2 \oplus CH_{share} \tag{24}$$

ii. CH checks if $T_4 - T_3 < \theta$. If true, the session continues; otherwise, it terminates.

iii. CH computes:

$$\varphi_1^* = \gamma_3 \oplus h(CH_{EA}\|r_4\|r_3) \tag{25}$$

$$\gamma_1^* = h\left(CH_{EA}\|\varphi_1^*\|CH_{share}\|r_4\|T_3\right) \tag{26}$$

iv. If $\gamma_1 = \gamma_1^*$, the values are verified. Otherwise, the session terminates.

v. CH generates a random number $r_5$ and computes:

$$\omega = h\left(\varphi_1^*\|r_3\|r_4\|r_5\right),$$
$$\rho_1 = h(T_4\|r_5\|CH_{EA}\|CH_{share}\|T_3\|\omega), \tag{27}$$
$$\rho_2 = h(r_5\|T_4) \oplus r_4$$

vi. CH transmits $\omega, \rho_1, \rho_2$ to the $\mathcal{G}$.

### 3. Steps Executed by the User Device

i. The user device receives $\gamma_1, \sigma_1,$ and $\sigma_2$ at timestamp $T_6$ and computes:

$$(r_5\|r_4\|T_5) = \sigma_2 \oplus r_3 \tag{28}$$

ii. It checks if $T_6 - T_5 < \theta$. If true, the session continues; otherwise, it terminates.

iii. The user device computes:

$$\sigma_1^* = h(\omega\|\varphi_1\|r_4\|T_5\|\gamma_1) \tag{29}$$

iv. If $\sigma_1 = \sigma_1^*$, $\mathcal{G}$ and CH are verified. Otherwise, the session terminates.

The summary of the mutual authentication phase is provided in Fig. 4. All cryptographic key exchanges and authentication sessions are securely logged on the blockchain for accountability and auditability.

### User password replacement phase

The user password replacement phase ensures secure updates of passwords for stakeholders' devices. The following are the stages involved in this phase:

Stage 1: On the reception of the logging details $\beta_2$, $\beta_3$, the

Stage 1: The user enters $SH_{id}$ and $SH_{psw}$ and requests to replace the password.

Stage 2: The user selects a new password $SH_{psw}^*$

Stage 3: The mobile device then computes the following values:

$$r_1^* = E_{id} \oplus h\left(SH_{psw}\|SH_{id}\right) \tag{30}$$

$$E_{psw}^* = h\left(r_1^* \oplus SH_{psw} \oplus d_i\right) \tag{31}$$

$$E_{psw}^* = h\left(r_1 \oplus SH_{psw}^* \oplus d_i\right) \tag{32}$$

Stage 4: The device sends $E_{psw}^*$ and the request for password replacement to $\mathcal{G}$.

# PeerJ Computer Science

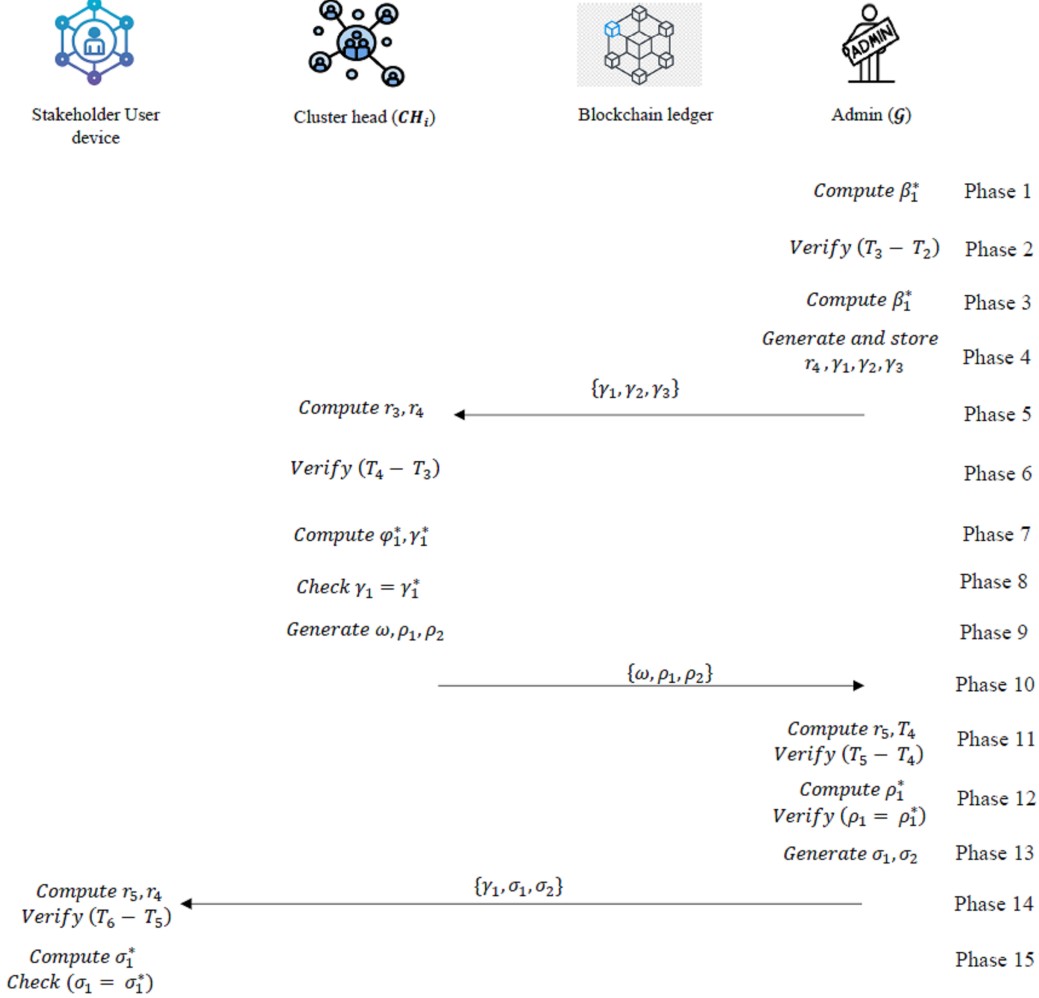

| Stakeholder User device | Cluster head ($CH_i$) | Blockchain ledger | Admin ($\mathcal{G}$) | |
|---|---|---|---|---|
| | | | Compute $\beta_1^*$ | Phase 1 |
| | | | Verify $(T_3 - T_2)$ | Phase 2 |
| | | | Compute $\beta_1^*$ | Phase 3 |
| | | | Generate and store $r_4, \gamma_1, \gamma_2, \gamma_3$ | Phase 4 |
| Compute $r_3, r_4$ | $\{\gamma_1, \gamma_2, \gamma_3\}$ | | | Phase 5 |
| Verify $(T_4 - T_3)$ | | | | Phase 6 |
| Compute $\varphi_1^*, \gamma_1^*$ | | | | Phase 7 |
| Check $\gamma_1 = \gamma_1^*$ | | | | Phase 8 |
| Generate $\omega, \rho_1, \rho_2$ | | | | Phase 9 |
| | $\{\omega, \rho_1, \rho_2\}$ | | | Phase 10 |
| | | | Compute $r_5, T_4$ Verify $(T_5 - T_4)$ | Phase 11 |
| | | | Compute $\rho_1^*$ Verify $(\rho_1 = \rho_1^*)$ | Phase 12 |
| | | | Generate $\sigma_1, \sigma_2$ | Phase 13 |
| Compute $r_5, r_4$ Verify $(T_6 - T_5)$ | $\{\gamma_1, \sigma_1, \sigma_2\}$ | | | Phase 14 |
| Compute $\sigma_1^*$ Check $(\sigma_1 = \sigma_1^*)$ | | | | Phase 15 |

**Figure 4 Overview of the mutual authentication phase.**

Stage 5: $\mathcal{G}$ computes:

$$\varphi_2^* = h\left(E_{psw}^*\|\mathcal{G}_{EA}\right) \tag{33}$$

$$\varphi_1^* = h\left(E_{psw}^*\|T_1\right) \tag{34}$$

$$\varphi_3^* = h\left(\varphi_1^*\|r_2\|\mathcal{G}_{EA}\right) \oplus h\left(E_{psw}^*\|T_1\right) \tag{35}$$

These values ($\varphi_2^*$, $\varphi_1^*$, $\varphi_3^*$) are shared with the mobile device.

Stage 6: The mobile device stores these values. With this, the old password is replaced successfully.

The summary of the user password replacement phase is provided in Fig. 5.

Following mutual authentication between stakeholders and related devices, the framework advances to a second layer of security controlling data access authorisation and the protection of sensitive information. This layer guarantees that private information is never leaked and that only authorised persons can access certain resources.

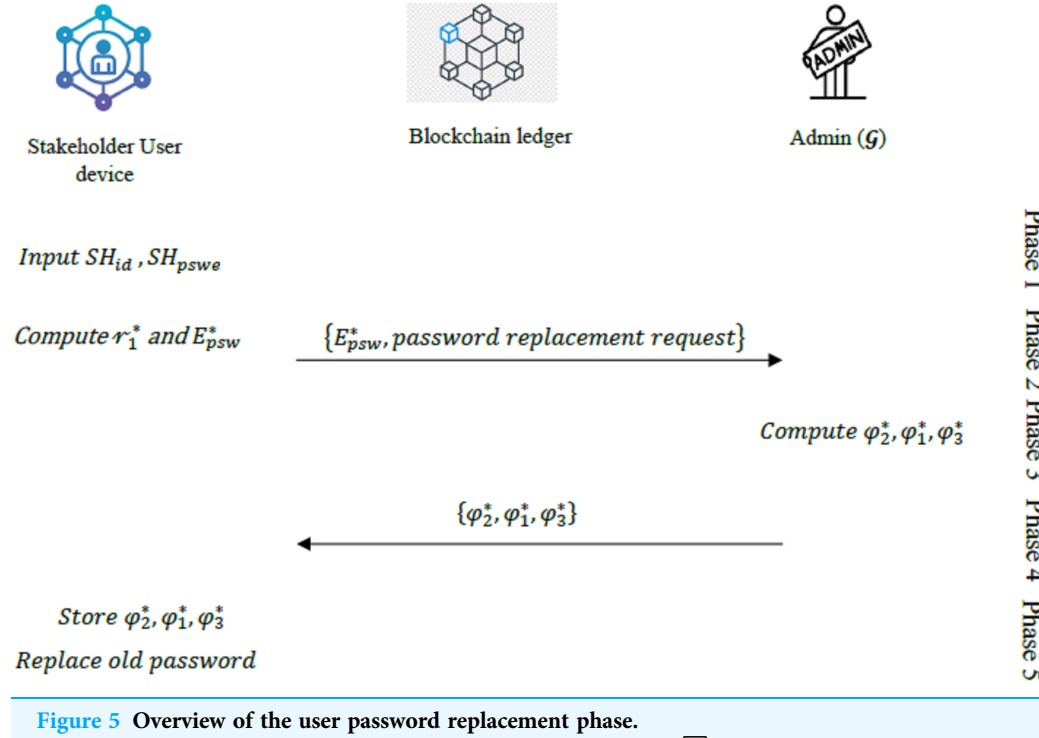

Stakeholder User device

Blockchain ledger

Admin ($\mathcal{G}$)

Input $SH_{id}$, $SH_{pswe}$

Compute $r_1^*$ and $E_{psw}^*$

$\{E_{psw}^*, password\ replacement\ request\}$

Compute $\varphi_2^*, \varphi_1^*, \varphi_3^*$

$\{\varphi_2^*, \varphi_1^*, \varphi_3^*\}$

Store $\varphi_2^*, \varphi_1^*, \varphi_3^*$

Replace old password

Phase 1    Phase 2    Phase 3    Phase 4    Phase 5

**Figure 5** **Overview of the user password replacement phase.**

## Second security level: security considerations during interactions among stakeholders in the framework

The RiceChain-Plus framework has been designed to address the critical security and privacy requirements of a blockchain-based rice supply chain system. This section demonstrates how the proposed system achieves Authorization and Privacy requirements during several interactions.

### Authorization

The RiceChain-Plus architecture employs a hybrid Role-Based Access Control (RBAC) and Attribute-Based Access Control (ABAC) approach to guarantee that only authorised stakeholders can access and alter certain data inside the rice supply chain. This is essential because of the system's multi-stakeholder composition, whereby entities such as farmers, processors, distributors, retailers, and consumers need regulated access to supply chain information. The hybrid RBAC and ABAC methods were chosen for their adaptability and precision in managing permissions across various stakeholder roles.

*i. Role-Based Access Control (RBAC):*

In this work, several stakeholders own unique roles and responsibilities, with their access to data or functions defined by their specific role. RBAC in RiceChain-Plus aligns the system's responsibilities with a predetermined array of permissions, so guaranteeing that stakeholders can access only the resources pertinent to their function.

(a) *Roles (R):* The proposed framework defines various roles for participants in the rice supply chain:

$$R = \{Admin, Supplier\ (SS), Farmer\ (FL), Grain\ Elevator\ (GE), Processor\ (GP),$$

$$Distributor\ (GD), Retailer\ (GR), Consumer\ (GC)\} \tag{36}$$

- *Admin:* Has full control over the system and manages permissions.
- *Supplier (SS):* Provides seeds and inputs and has access to data related to supplies.
- *Farmer (FL):* Cultivates the crops and inputs data regarding farm conditions.
- *Grain Elevator (GE):* Manages post-harvest storage and transport.
- *Processor (GP):* Handles the processing and transformation of rice.
- *Distributor (GD):* Distributes rice to retailers and consumers.
- *Retailer (GR):* Manages retail sales and consumer transactions.
- *Consumer (GC):* Purchases and consumes the rice.

(b) *Permissions (P):* Each role is assigned a specific set of permissions:

$$P = \{Read, Write, Update, Delete\} \tag{37}$$

A farmer may be authorised to document data on rice cultivation (*e.g.*, seed variety, fertiliser use) but may lack the authority to erase processing records. A Processor may possess the authority to change or amend the status of processed rice, although lacks the ability to remove data pertaining to transactions between retailers and consumers.

(c) *Access Control List (ACL):* The ACL maps roles to their permissions:

$$ACL : R \rightarrow 2^P \tag{38}$$

For example:

$$ACL(Farmer) = \{Write, Update\}$$

$$ACL(Processor) = \{Read, Update, Write\}$$

This guarantees that stakeholders can only view data and execute operations relevant to their roles within the rice supply chain.

ii. *Attribute-Based Access Control (ABAC)*

In this work, the RBAC regulates access according to roles, but ABAC improves the authorisation process by integrating characteristics associated with the user, data, and environment. This facilitates more precise access control by considering supplementary parameters such as data sensitivity, geographical location, and temporal context.

(a) *Attributes (A):* Here, attributes can be related to the user, data type, or environmental conditions:

$$A = \{Role, Time, Location, Data\ Type, Data\ Sensitivity\} \tag{39}$$

Examples of characteristics encompass:

- *Temporal Restrictions:* Access can be limited according to the time of day or operating hours.

- *Location:* Specific stakeholders may access just data relevant to their geographical region. A farmer in a particular area can be unable to access data from other regions.
- *Data Sensitivity:* Consumer information is sensitive and can only be accessed by authorised individuals, such as Retailers or Administrators.

(b) *Access Request (AR):* Each user request is assessed according to its characteristics associated with both the user and the data being accessed. This can be expressed as:

$$AR = (u, \ d, \ act) \tag{40}$$

where:

- $u$ is the requesting user,
- $d$ is the data object (*e.g.*, transaction record, farm report),
- $act$ is the requested action (*e.g.*, Read, Write).

(c) *Attribute Mapping (AM):* Before access is authorised, each user, data object, or environment has set of characteristics that need evaluation:

$$AM{:}U \cup Data \cup Environment \rightarrow 2^A \tag{41}$$

For example, a Processor can assign attributes like:

$$A_u = \{Role{:}Processor, Location{:}Processing \ Plant, Data \ Type{:}Processing \ Record, Data$$

$$Sensitivity{:}Medium\} \tag{42}$$

(d) *Policy Evaluation (PE)*

To decide whether access should be permitted, the Policy Evaluation function examines both the attribute-based conditions and the role-based rights. The system grades an access request as follows:

$$PE{:}AR \times P \rightarrow \{true, false\} \tag{43}$$

$$PE(u, d, act) = \begin{cases} true & if \ Role(u) \in R \land act \in PA(Role(u)) \land \forall \ a \in A_u, \\ & attribute\_check(a, d, act) \\ fale & otherwise \end{cases} \tag{44}$$

This approach guarantees that access is allowed only in cases of:

- The user's role allows the desired action,
- And user's characteristics match those of the data, including time, location, and data sensitivity.

For example, if a given Retailer $u$ request to Read a transaction record $d$, before granting him, the system will check:

- If the Retailer's role permits the action "Read".
- If its characteristics matches with the request.

If both conditions are satisfied, access is granted; otherwise, it is denied.

The RiceChain-Plus architecture adaptively modifies authorisation rules according on stakeholder roles and data sensitivity. The administrative entity oversees the policy update process, validating all access modifications using cryptographic signatures. All changes are recorded on-chain, guaranteeing openness and auditability of role modifications. Routine policy audits are conducted to avert privilege escalation attacks.

### Privacy

The proposed framework guarantees the privacy of confidential information by employing Zero-Knowledge Proofs (ZKPs). These cryptographic methods ensure that privacy is maintained all through the supply chain by letting stakeholders check important data without disclosing the underlying sensitive information. ZKPs specifically allow stakeholders to demonstrate ownership or authenticity of certain data (like transaction data, location, or quality measurements) without disclosing the real values or data content.

ZKPs are incorporated at various levels where sensitive data has to be validated throughout the supply chain without revealing the real data, therefore guaranteeing privacy under the proposed framework. In this framework, ZKP design proceeds in following phases:

(a) *Phase-1: Proof Generation*

Let $M$ be the secret message representing sensitive information (*e.g.*, the batch quality of rice, the financial information of a transaction, or a precise GPS position of a storage facility). The prover—a stakeholder—such as a retailer or processor—wishes to show the verifier—another stakeholder—that the data $M$ satisfies certain requirements. $P(M)$ (such as demonstrating that the rice batch conforms with the established quality criteria) without disclosing $M$. This can be represented as follow:

$$\pi = Prove(P(M)) \tag{45}$$

where $\pi$ is the zero-knowledge proof generated by the prover based on the secret message $M$ and the statement $P(M)$.

For example, a given processor may wish to certify that a batch of rice satisfies quality criteria without exposing its composition or processing. Under this situation, $P(M)$ can represent a claim like "the rice moisture content is within acceptable limits." Based on this assumption, the processor can offer a proof $\pi$ without disclosing the real moisture content.

(b) *Phase-2: Proof Verification*

On receiving the proof from the prover, the verifier confirms it with the public statement $P(M)$. ZKPs are unique in such a way that the verifier can verify the accuracy of the proof without learning the secret message $M$ as follows:

$$Verify(\pi, P(M)) = true\ or\ false \tag{46}$$

Should the proof be valid, the verifier treats it as accurate; otherwise, it is denied. This guarantees, in this context, that participants may confirm claims made by others (*e.g.*, quality criteria or transaction legitimacy) without directly access to private data.

(c) *Phase-3: Integration with Blockchain*

The proposed framework guarantees privacy by making sure ZKPs are safely logged on the blockchain without disclosing private information. this guarantees that:

- Proofs are verifiable and unchangeable.
- Sensitive information—such as transaction records or quality standards—is kept private.
- Responsibility is upheld without invading personal space.

The blockchain merely logs the proofs and verification results, therefore assuring that no sensitive information is revealed even when the network can confirm compliance or accuracy.

### Summary of the proposed model

In summary, the RiceChain-Plus framework secures transaction flows using a two-level security model that ensures robust authentication, authorization, and privacy preservation across the rice supply chain. Below is the transaction flow summary:

(1) *Authenticate User & Device:* Secure login with hashed credentials, nonces, and timestamps.
(2) *Mutual Authentication:* Both user and system entities verify each other to ensure trust.
(3) *Access Control:* RBAC and ABAC determine what data or functions the user can access.
(4) *Data Privacy Verification:* ZKPs ensure sensitive data is validated without exposure.
(5) *Transaction Execution:* Verified transactions are recorded on the blockchain for immutability and traceability.

To encapsulate the flow of interactions in the proposed framework, Algorithm 1 and Fig. 6 offer a procedural overview of the RiceChain-Plus integration pipeline.

## SECURITY ANALYSIS

The effectiveness of the proposed model against side channel, massage and identity, and replay attacks is evaluated by a comprehensive security analysis. In addition, a BAN logic-based security analysis of the proposed model was conducted to assess its effectiveness in achieving the desired authentication goals. The following subsections provides detailed discussion on the security efficiency of the proposed model.

### Mitigation strategies for internal threats

The RiceChain-Plus architecture utilises strong methods to protect against internal threats, including malicious insiders (*e.g.*, dishonest stakeholders like farmers or processors) and compromised devices (*e.g.*, infiltrated sensor nodes or cluster heads). These countermeasures guarantee data integrity, message authenticity, and comprehensive resistance against threats, including replay attacks.

| | **Algorithm 1** RiceChain-Plus framework integration. |
|---|---|
| 1 | def transaction_flow(user, device, data): |
| 2 | # *First Security Level: Authentication* |
| 3 | if not authenticate_user(user, device): |
| 4 | return "Authentication Failed" |
| 5 | # *Mutual Authentication* |
| 6 | if not mutual_authentication(user, device): |
| 7 | return "Mutual Authentication Failed" |
| 8 | # *Second Security Level: Authorization* |
| 9 | if not authorize_access(user, data): |
| 10 | return "Access Denied" |
| 12 | # *Privacy Preservation with ZKP* |
| 13 | zk_proof = generate_zkp(data) |
| 14 | if not verify_zkp(zk_proof): |
| 15 | return "ZKP Verification Failed" |
| 16 | # *Transaction Execution* |
| 17 | transaction = execute_transaction(user, data) |
| 18 | log_transaction(transaction) |
| 19 | return "Transaction Successful" |
| 20 | def authenticate_user(user, device): |
| 21 | # *Validate user credentials and device* |
| 22 | return validate_credentials(user) and verify_device(device) |
| 23 | def mutual_authentication(user, device): |
| 24 | # *Use nonces and timestamps to prevent replay attacks* |
| 25 | nonce = generate_nonce() |
| 26 | timestamp = current_time() |
| 27 | return validate_nonce(nonce) and validate_timestamp(timestamp) |
| 28 | def authorize_access(user, data): |
| 29 | # *Evaluate RBAC and ABAC policies* |
| 30 | return check_rbac(user.role, data) and check_abac(user.attributes, data) |
| 31 | def generate_zkp(data): |
| 32 | # *Create Zero-Knowledge Proof for data verification* |
| 33 | return zk_proof_algorithm(data) |
| 34 | def verify_zkp(zk_proof): |
| 35 | # *Verify the ZKP without revealing the data* |
| 36 | return zk_verifier(zk_proof) |
| 37 | def execute_transaction(user, data): |
| 38 | # *Execute and log the transaction on the blockchain* |

| | Algorithm 1 (continued) |
|---|---|
| 39 | return blockchain_execute(user, data) |
| 40 | def log_transaction(transaction): |
| 41 | # Log transaction details securely |
| 42 | blockchain_log(transaction) |

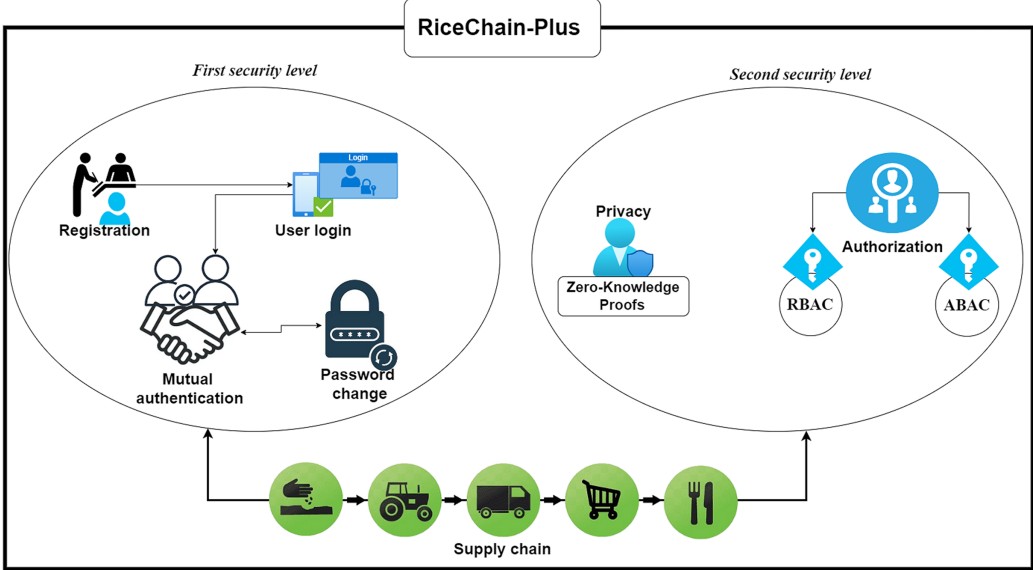

**Figure 6 Overview of the proposed framework integration.** This figure shows RiceChain-Plus, a proposed framework integration for the rice supply chain, integrating blockchain technology, cryptographic security mechanisms, and access control models. Key components include stakeholder registration, transaction validation, data authentication, and decentralized ledger interactions for transparency and security.

(a) *Safeguarding Against Malicious Insiders*

Malicious insiders may seek to manipulate data, fabricate transactions, or interfere with the supply chain. RiceChain-Plus mitigates these vulnerabilities by using cryptographic hash methods, ensuring blockchain immutability, and implementing defences against replay attacks:

   i. *Ensuring Data Integrity using Hashing:*

- The architecture uses real-time validation at fog nodes to ensure data legitimacy, with each packet passing cryptographic integrity tests before being sent to the blockchain.
- Sensor nodes have distinct IDs and session-specific nonces, ensuring data from authorized devices is permitted, thus preventing tampering or erroneous data during data gathering.

- All interactions, transactions, or data blocks are hashed using the SHA-256 cryptographic hash algorithm, guaranteeing that any alteration to the data produces a unique hash result.
- For instance, the framework calculates a hash $h(M) = SHA256(M)$ for a transaction that includes sensitive data (*e.g.*, rice quality measurements). The hash value is recorded on the blockchain in conjunction with the transaction data.
- During verification, the incoming data is rehashed, and the computed hash is compared to the stored hash. Any discrepancy signifies manipulation, resulting in the failure of the integrity check and obstructing malicious insiders from modifying data undetected.

ii. *Irreversibility of Transactions:*

- Once a transaction is inscribed in the blockchain, it becomes immutable, rendering it hard for malicious insiders to alter or erase data without notice.
- This guarantees responsibility for all parties, since any effort to falsify or alter previous data is impracticable.

iii. *Protection Against Replay Attacks:*

- Replay attacks include the interception of authentic data and its subsequent malicious retransmission to get unauthorised access or perpetrate fraud. RiceChain-Plus safeguards against such attempts using timestamps and nonces:
  - *Timestamp:* Each transaction includes a timestamp (*e.g.*, $T_1$) to guarantee data validity within a designated period. Upon receipt of a message, the system verifies the current time ($T_2$) and computes the difference $T_2 - T_1$. If this surpasses the maximum permissible delay ($\theta$), the session is ended, and the message is designated as stale. This inhibits adversaries from retransmitting obsolete communications or transactions.
  - *Nonces:* Randomly generated numbers for each session or transaction guarantee uniqueness. A nonce that has been previously used is never allowed again, rendering the reuse of intercepted transactions in a replay attack infeasible.
- In the process of mutual authentication, the mobile device produces a session identification $\beta_2$ using a random nonce $r_3$. The admin gateway $\mathcal{G}$ checks both $r_3$ and the timestamp $T_2$. Upon detection of a duplicate nonce or an expired timestamp, the session is terminated, therefore thwarting replay attempts.

iv. *Mitigation of Fabricated Communications:*

- Mutual authentication systems prevent malicious insiders from producing legitimate communications to impersonate another party. To generate a counterfeit message $\{d_i, \beta_2, \beta_3\}$, the insider must calculate the following parameters:

$$\beta_2 = h(T_2 \| r_3 \| \beta_1 \| \mathcal{G}_{EA})$$

$$\beta_3 = (r_3 \| T_2) \oplus \beta_1.$$

- Nonetheless, in the absence of access to $\beta_1$, contingent upon private credentials like $SH_{psw}$ (stakeholder password) and $r_1$ (random number), the insider is unable to generate correct values. These credentials are safeguarded by one-way hash algorithms and cannot be inferred or replicated within polynomial time.

(b) *Protection Against Compromised Devices*

Infiltrated sensor nodes or cluster heads may transmit erroneous data to interrupt the supply chain or alter blockchain records. RiceChain-Plus alleviates this threat by means of:

   *i. Validation of Data Authenticity:*

- Authentication of each device's communication occurs prior to the recording of its data on the blockchain. For example, communications across sensor nodes contain cryptographic hashes $\varphi_1$, $\varphi_1$, $\beta_1$, and session-specific random numbers $r_3, r_4$, which are authenticated by the cluster head or administrative gateway $\mathcal{G}$.
- Should any parameter fail validation, the data is discarded, therefore preventing exploited devices from introducing erroneous information.

   *ii. Tamper Resistance using Mutual Authentication:*

- Mutual authentication guarantees that only permitted devices may engage in the protocol. For a compromised cluster head to successfully fabricate a legitimate message, it must fulfil the following criteria:

$$\beta_2 = h(T_2 \| r_3 \| \beta_1 \| \mathcal{G}_{EA})$$

$$\beta_3 = (r_3 \| T_2) \oplus \beta_1.$$

- In the absence of session-specific parameters such as $\beta_1$, $r_3$, and $r_4$, the compromised device is incapable of producing authentic messages.

   *iii. Resistance to Replay Attacks:*

- Replay attack mitigation strategies are also applicable to compromised devices. Timestamps and nonces guarantee that intercepted legitimate data cannot be repurposed to fabricate new transactions. Any attempt to retransmit previous communications is identified and denied by the verification procedure.

   *iv. Integrity of Recorded Data:*

Despite a compromised device transmitting erroneous data, the verification procedure guarantees the preservation of data integrity. Only data that passes integrity tests is kept on the blockchain. Efforts to alter current records will be unsuccessful because of blockchain immutability.

The RiceChain-Plus architecture offers extensive safeguards against internal threats by guaranteeing data integrity, message authenticity, avoidance of replay attacks, and

stringent device validation. The system efficiently eliminates data tampering, transaction forgery, and supply chain disruptions by using cryptographic hashes, timestamps, nonces, mutual authentication, and the immutability of blockchain technology. These procedures guarantee the dependability and safety of the supply chain, even against internal challenges.

## Resilience against external adversaries

The proposed framework exhibits significant resistance against external attackers targeting related devices, including sensor nodes and gateway nodes. This resilience is achieved by strong cryptographic techniques and the secure management of protocol parameters, as shown below:

*Defence Against Adversarial Communication Eavesdropping*: The architecture presupposes that an adversary could intercept communications during the execution of the protocol. An adversary can attempt to formulate a legitimate message for $\mathcal{G}$ after capturing the parameters $\rho_1$ and $\rho_2$, where:

$$\rho_1 = h(T_4 \| r_5 \| CH_{EA} \| CH_{share} \| T_3 \| \omega),$$
$$\rho_2 = h(r_5 \| T_4) \oplus r_4$$

Nevertheless, the adversary is unable to produce a legitimate message without access to the secret values $\omega$ and $r_5$, which are safeguarded by a one-way hash function. The irreversible characteristic of the hash function guarantees that these values cannot be rebuilt from captured data. Consequently, the adversary is unable to extract significant information or generate authentic messages, hence guaranteeing the protocol's resilience against side-channel attacks aimed at sensor nodes.

*Validation of Gateway Node Authenticity*: The system guarantees that sensor nodes and user devices can authenticate the gateway node's legitimacy in two principal attack scenarios:

1. *Intercepted Communication Between $\mathcal{G}$ and Sensor Node:*

Should an adversary intercept the communication $\{CH_{EA}, \gamma_1, \gamma_2, \gamma_3\}$ over the public channel, they may attempt to impersonate the gateway. Nonetheless:

- $\gamma_1, \gamma_2, \gamma_3$ are contingent upon the random integer $r_4$, which is secured by the one-way hash algorithm $h()$. The irreversible characteristic of $h()$ inhibits an adversary from deriving $r_4$.
- $CH_{share}$ is a cryptographic parameter that cannot be deduced in polynomial time because of its robust security.

Collectively, these procedures guarantee that the gateway node remains unimpersonated.

2. *Intercepted Communication Between $\mathcal{G}$ and User Device:*

If an adversary intercepts the message $\{\gamma_1, \sigma_1, \sigma_2\}$, they may attempt to impersonate the gateway. Nonetheless:

- In the absence of $\varphi_1$ and $r_4$, the adversary is unable to calculate $\sigma_1$ or $\sigma_2$.

- The irreversibility of the one-way hash function guarantees that $\varphi_1$ and $r_4$ cannot be rebuilt from intercepted data.

Thus, when the user device identifies an incorrect message, it ends the connection, thus mitigating gateway simulation attacks.

The proposed system utilises cryptographic primitives, including one-way hash functions and secure key exchange, to provide strong protection against external threats aimed at sensor and gateway nodes. These techniques protect communication, inhibit unauthorised access, and obstruct impersonation attempts, preserving the integrity and security of the supply chain network.

## Proof of correctness for mutual authentication and key exchange phase

To verify the accuracy of the proposed mutual authentication protocol, we use the Burrows-Abadi-Needham (BAN) logic, a recognised framework for assessing the security attributes of authentication and key exchange protocols. The analysis confirms that the protocol attains the requisite security attributes, including mutual authentication, key freshness, and key validity. The following sections analyse the mutual authentication and key exchange phase of the proposed model using BAN logic to ensure it conforms to the desired security properties.

*Key Concepts in BAN Logic*

(1) *Beliefs:* What principals (*e.g.*, admin $\mathcal{G}$ and cluster head CH) believe to be true.

(2) *Freshness:* A statement or a key is fresh if it has not been used before.

(3) *Jurisdiction:* A principal can be trusted to say something about another principal.

(4) *Nonce:* A random number used only once within a protocol run.

   *BAN Logic Notations*

- $P|\equiv X$: Principal $P$ believes $X$.
- $P \triangleleft X$: Principal $P$ sees $X$.
- $P|\sim X$ : Principal $P$ said $X$.
- $P|\Rightarrow X$: Principal $P$ has jurisdiction over $X$.
- $\#(X)$: $X$ is fresh.

   *Protocol Elements*

- Principals: $\mathcal{G}$ and CH.
- Shared Keys: $\mathcal{G}_{skey}$ (shared key between $\mathcal{G}$ and CH), $CH_{share}$ (key derived by CH).
- Timestamps: $T_3, T_4, T_5$, representing specific steps in the protocol.
- Random Numbers: $r_3, r_4, r_5$, these ensures freshness of each session.

   Initial Assumptions

(1) $\mathcal{G}|\equiv \#(r_4)$: $\mathcal{G}$ believes $r_4$ is fresh.

(2) $\mathcal{G}|\equiv \mathcal{G}_{skey}$: $\mathcal{G}$ believes $\mathcal{G}_{skey}$ is a good, shared key.

(3) $\mathcal{G}|\equiv \mathcal{G}_{skey} \Rightarrow \{CH|\equiv \mathcal{G}_{skey}\}$: $\mathcal{G}$ trusts $CH$ to know $\mathcal{G}_{skey}$.

*Protocol Messages and Translations*

*Message 1: $\mathcal{G} \rightarrow CH$: $\gamma_1, \gamma_2, \gamma_3$*

- $\mathcal{G}| \equiv \#(T_3)$: $\mathcal{G}$ believes $T_3$ is fresh.
- $\mathcal{G}| \equiv \#(r_4)$: $\mathcal{G}$ believes $r_4$ is fresh.
- $CH \lhd \gamma_1, \gamma_2, \gamma_3$: $CH$ sees $\gamma_1, \gamma_2, \gamma_3$.
- From $\gamma_1 = h(CH_{EA}\|\varphi_1\|CH_{share}\|r_4\|T_3)$:
  - $CH| \equiv \#(T_3)$: CH believes $T_3$ is fresh.
  - $CH| \equiv \#(r_4)$: CH believes $r_4$ is fresh.
- $CH| \equiv \mathcal{G}| \equiv \mathcal{G}_{skey} \Rightarrow \{CH| \equiv \mathcal{G}_{skey}\}$:CH believes $\mathcal{G}$.
- $CH| \equiv \mathcal{G}_{skey}$:CH believes the shared key $\mathcal{G}_{skey}$ is valid.

*Message 2: CH $\rightarrow \mathcal{G}$: $\omega, \rho_1, \rho_2$*

- $CH | \equiv \#(T_4)$: $CH$ believes $T_4$ is fresh.
- $CH | \equiv \#(r_5)$: $CH$ believes $r_5$ is fresh.
- $\mathcal{G} \lhd \omega, \rho_1, \rho_2$: $\mathcal{G}$ sees $\omega, \rho_1, \rho_2$.
- From $\rho_1 = h(T_4\|r_5\|CH_{EA}\|CH_{share}\|T_3\|\omega)$:

  - $\mathcal{G}| \equiv \#(T_4)$: $\mathcal{G}$ believes 4 is fresh.
  - $\mathcal{G}| \equiv \#(r_5)$: $\mathcal{G}$ believes $r_5$ is fresh.

- $\mathcal{G}| \equiv CH| \equiv \mathcal{G}_{skey} \Rightarrow \{\mathcal{G} \equiv \mathcal{G}_{skey}\}$: $\mathcal{G}$ believes $CH$ trust $\mathcal{G}$.
- $\mathcal{G}| \equiv \mathcal{G}_{skey}$: $\mathcal{G}$ believes the shared key $\mathcal{G}_{skey}$ is valid.

*Goals Achieved*

(1) $\mathcal{G}| \equiv CH| \equiv \mathcal{G}_{skey}$: $\mathcal{G}$ believes $CH$ trust the shared key $\mathcal{G}_{skey}$.

(2) $CH| \equiv \mathcal{G}| \equiv \mathcal{G}_{skey}$: CH believes $\mathcal{G}$ trust the shared key $\mathcal{G}_{skey}$.

The proposed mutual authentication and key exchange procedure adheres to the security attributes mandated by BAN logic. The analysis guarantees mutual authentication between the admin $\mathcal{G}$ and the cluster head $CH$, as well as the confidence in the established shared key $\mathcal{G}_{skey}$. The system ensures key freshness, safeguards against replay attacks, and facilitates secure mutual authentication. This formal proof of correctness confirms the resilience of the authentication step against prevalent threats in supply chain systems. To complement the security analysis, we conducted a detailed performance evaluation of the RiceChain-Plus framework under simulated network conditions, as described in the next section.

## PERFORMANCE EVALUATION AND DISCUSSIONS

This section focuses on the experimental conditions and performance criteria used to assess the effectiveness of the RiceChain-Plus model. This practical approach enables us to optimize the framework, address any unanticipated problems, and guarantee its resilience

and effectiveness when used in real-world agricultural supply chains. A comparison with MBRRSM (2022) (*Peng et al., 2022b*), KRanTi (2024) (*Patel et al., 2024*), PoTx (2023) (*Saranya & Maheswari, 2023*), and RiceChain (2022) (*Yakubu et al., 2022*) was done to see how well the proposed model would work with better blockchain technology for agricultural supply chains. MBRRSM combines multiple blockchains and enhanced cryptographic methods, making it appropriate for evaluating our single blockchain approach using PoA consensus. KRanTi uses the Ethereum blockchain and IPFS to provide optimal data storage and retrieval, thereby establishing a standard for our storage efficiency. PoTx primarily aims to enhance transaction verification and minimize energy usage, providing a meaningful benchmark for evaluating the efficiency of our PoA-based methodology. RiceChain (2022), our predecessor, functions as a benchmark to emphasize the innovations and enhancements in RiceChain-Plus.

## Simulation setup

The simulation was conducted using computers with Intel Core i7-6700 CPUs at 3.40 GHz, implementing the RiceChain-Plus model in Solidity and interacting with the Ethereum Virtual Machine (EVM) *via* Python modules like Web3.py. The simulation used the Metamask Ethereum wallet, the Rinkeby testnet, and a Proof-of-Authority (PoA) consensus method, ensuring uniform and unbiased testing conditions. Each scenario involved 100 transaction interactions between stakeholders over 100 min, allowing for activities like mutual authentication. This is based on accepted benchmarking approaches for assessing blockchain-based supply chains. This method offers statistically notable analysis of system performance. Although further extensive testing is expected for future work, the present findings satisfactorily support the efficiency of the system.

The simulation generated comprehensive data, enabling rigorous monitoring and validation of the RiceChain-Plus smart contract. The smart contract's security was assessed using the Oyente tool, achieving 68% coverage of the EVM code and confirming the absence of common security flaws. Table 4 presents the smart contract vulnerability analysis for RiceChain-Plus, demonstrating its resilience against common blockchain attacks. The framework effectively mitigates risks such as reentrancy, integer overflows, and unauthorized access through well-defined security measures. The inclusion of role-based access control, nonces, and timestamp validation ensures robust protection against malicious exploits.

RiceChain-Plus has a cluster-based design in which each cluster head (CH) gathers and authenticates data prior to transmitting it to the blockchain. The model has not yet been implemented in a practical environment, but its design incorporates IoT devices with optimised resource allocation for enhanced energy efficiency. The frequency of data collection is continuously regulated according to network circumstances and stakeholder interactions to optimise real-time monitoring and computational efficiency. The measured transaction execution time in our simulations is around 44.5 ms, consistent with industry benchmarks for blockchain-based agricultural monitoring systems.

**Table 4 Smart contract vulnerability analysis.** Evaluation of the RiceChain-Plus smart contract's security, identifying and mitigating vulnerabilities such as integer overflow, underflow, transaction ordering dependence, reentrancy attacks, timestamp dependency, unauthorized access, DoS risks, and gas limit/loop attacks. The EVM code coverage measures 68%.

| Vulnerability type | Analysis result |
|---|---|
| Integer overflow | Not found |
| Integer underflow | Not found |
| Reentrancy attack | Mitigated with reentrancy guard |
| Transaction ordering dependence | Not found |
| Timestamp dependency | Handled with nonces & timestamps |
| Unauthorized access | Mitigated with role-based access control |
| Denial of service | Mitigated with rate-limiting |
| Gas limit and loop attacks | Optimized gas usage, no infinite loops |
| EVM code coverage | Coverage: 68% (Optimized code) |

In the simulation, each cluster head device was represented by a unique Ethereum account to mimic real-world scenarios. The computational cost of each device was compared to the gas usage using MetaMask, and the validity of the simulated gateway (admin) device was verified with a separate Ethereum account. The study tested all models based on a sequence of transactions, ensuring that accounts were adequately funded to avoid interruptions.

We must acknowledge that this study does not cover the potential consequences of a cluster head device running out of funds, which could be considered a limitation. Nevertheless, all accounts were funded throughout the system design to ensure uninterrupted functioning, thereby eliminating the possibility of devices running out of funds.

## Basic performance evaluations

The performance of RiceChain-Plus was assessed using several key criteria, including average execution costs, energy consumption, throughput, execution time, and transaction time. The results of these evaluations are detailed in the sections that follow.

### Gas cost analysis

Table 5 presents the cost analysis of Ethereum functions for RiceChain-Plus in terms of transaction and execution costs. The evaluation considers gas fees at an average price of 50 Gwei, with an exchange rate of $3,000 per ETH. The transaction with the highest cost was *mutualAuthenticate*, requiring 17,500 gas units, which translates to approximately $1.31 per execution. Meanwhile, other crucial functions such as *registerStakeholder* and *authenticateStakeholder* were optimized to cost $1.05 and $1.20 per execution, respectively.

These values indicate that RiceChain-Plus efficiently manages gas costs compared to standard Ethereum-based supply chain frameworks. The use of optimized cryptographic hashing and smart contract function modularization ensures that execution overhead is minimized, making the framework cost-effective and practical for large-scale deployment.

**Table 5 Gas cost analysis for RiceChain-Plus.** The RiceChain-Plus smart contract's Gas Cost Analysis shows that functions like mutual authentication and user authentication require higher computational resources, while registering devices and retrieving mobile details incur lower costs. The system maintains a balanced gas consumption profile for secure transactions.

| Function name | Transaction gas | Execution gas | Cost in USD |
|---|---|---|---|
| registerStakeholder | 14,000 | 9,000 | 6.3 |
| authenticateStakeholder | 16,000 | 11,000 | 7.19999999999999 |
| registerDevice | 13,500 | 8,500 | 6.074999999999999 |
| addClusterHead | 14,500 | 9,500 | 6.5249999999999995 |
| loginUser | 15,500 | 10,500 | 6.975 |
| mutualAuthenticate | 17,500 | 12,500 | 7.875000000000001 |
| replacePassword | 16,500 | 11,500 | 7.424999999999999 |
| getMobileDetails | 13,000 | 8,000 | 5.85 |

### Execution costs

This section compares the execution costs, specifically gas consumption, of the RiceChain-Plus model against four benchmark blockchain models: MBRRSM (2022), KRanTi (2024), PoTx (2023), and RiceChain (2022). RiceChain-Plus demonstrated the lowest and most consistent gas usage, ranging from approximately 43,860 to 45,407 gas units, indicating its high efficiency in transaction processing as depicted in Fig. 7. This efficiency is attributed to improvements in the consensus method, data management, and cryptographic approaches.

In contrast, the MBRRSM model had the highest gas consumption, suggesting it requires more computational resources due to complex or less optimized procedures. KRanTi showed moderate gas usage (46,800 to 48,277 gas units), affected by storage overheads and less optimal consensus mechanisms. PoTx had competitive gas consumption (42,800 to 44,205 gas units) due to its focus on optimizing transaction verification, but it still fell short of RiceChain-Plus's efficiency. RiceChain (2022) was efficient, with gas usage slightly higher than RiceChain-Plus, ranging from 43,800 to 45,277 gas units.

Overall, RiceChain-Plus consistently showed the lowest execution costs, confirming its superior efficiency and optimization, making it a cost-effective and viable solution for blockchain-based agricultural supply chains. The other models displayed higher and more variable gas consumption, indicating less efficient processes. This analysis underscores RiceChain-Plus's position as an efficient and practical choice for real-world applications.

### Energy consumption

RiceChain-Plus demonstrates superior energy efficiency of 9.38828E−05 J averagely compared to benchmark models MBRRSM (2022), KRanTi (2024), PoTx (2023), and RiceChain (2022) which have average energy consumption of 0.000112504, 9.99106E−05, 0.00009973 and 0.000101697 J respectively. This efficiency is consistently observed across various interactions, as depicted in the Fig. 8. The RiceChain-Plus model consumes significantly less energy, highlighting its effectiveness in energy utilization.

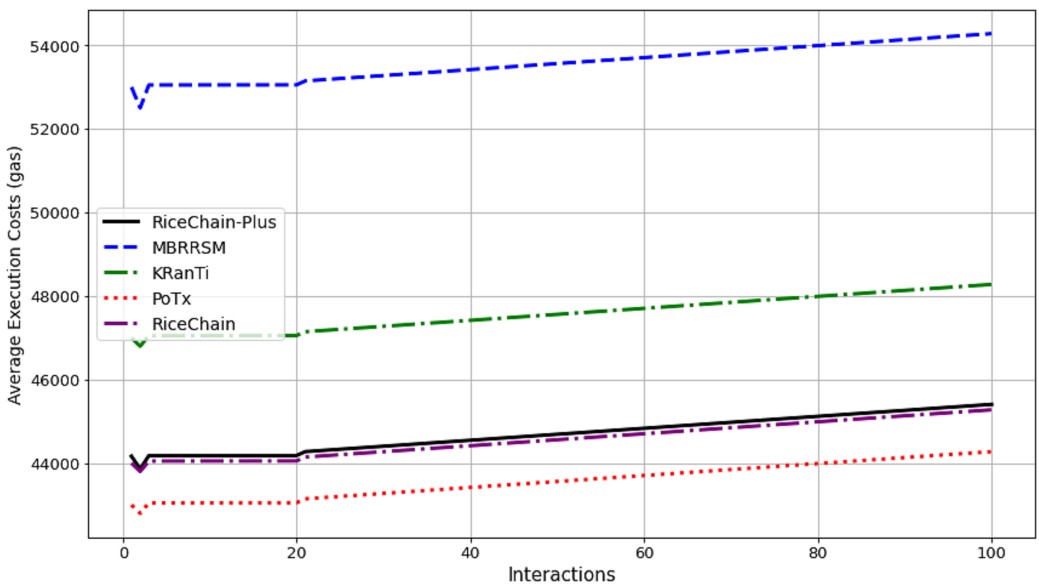

**Figure 7  Average execution costs across interactions.**

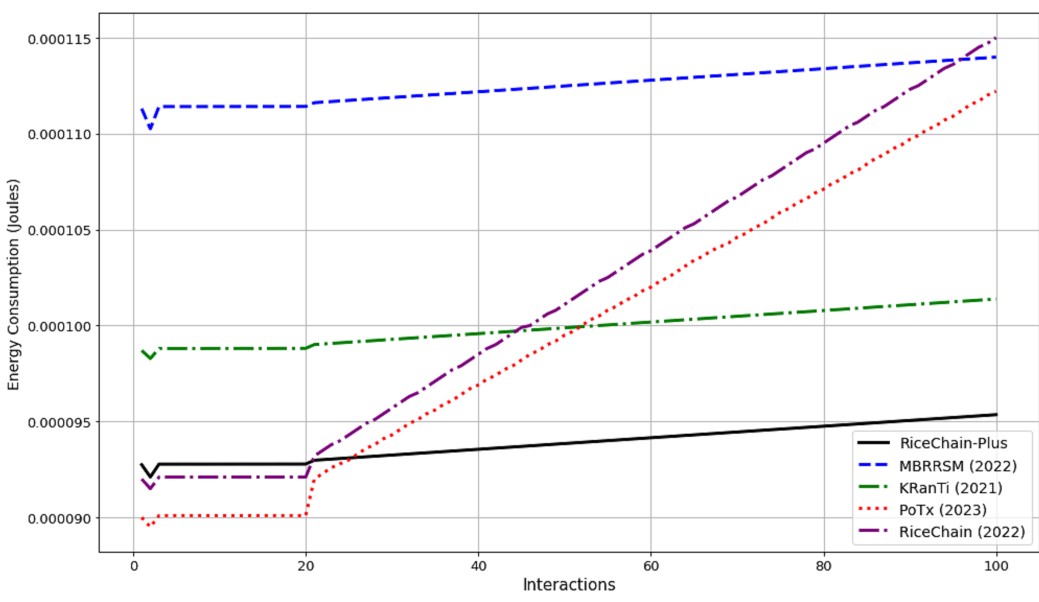

**Figure 8  Energy consumption comparison.**

The reduced energy consumption of RiceChain-Plus is largely attributed to its use of the PoA consensus mechanism, which is more energy-efficient than the Proof of Work (PoW) or Proof of Stake (PoS) mechanisms employed by the other models. Additionally, RiceChain-Plus leverages optimized SHA-256 hashing algorithms, which reduce computational overhead and minimize energy usage. This technological approach not only enhances energy efficiency but also speeds up transaction processing, contributing to the overall efficacy of the system.

The efficiency of RiceChain-Plus positively impacts its scalability and practical application, especially in agricultural supply chains where cost-efficiency and scalability are crucial. The model can handle a larger number of interactions without a significant increase in energy consumption, making it a cost-effective solution as the network grows. Although RiceChain-Plus has a slightly higher average execution cost compared to RiceChain (2022), its superior energy efficiency makes it the preferred option for managing agricultural supply chains.

### Throughput analysis

RiceChain-Plus consistently outperforms benchmark models in throughput, maintaining an average throughput of 0.071201 transactions per second, as illustrated in Fig. 9. The minimal variation between its highest value (0.071205493 at Interaction 2) and lowest value (0.071197653 at Interaction 100) indicates stable performance. In comparison, MBRRSM shows an average throughput of 0.064835, peaking at 0.065831238 during Interaction 3 and dipping to 0.063303417 at Interaction 100. KRanTi averages 0.064384, with a high of 0.064882949 at Interaction 2 and a low of 0.06247292 at Interaction 100. PoTx records an average throughput of 0.065127, reaching 0.066376943 at Interaction 3 and falling to 0.063787976 at Interaction 100. RiceChain's average is 0.064757, with values ranging from 0.06525077 at Interaction 2 to 0.062771114 at Interaction 100.

The superior throughput of RiceChain-Plus is attributed to its optimized PoA consensus mechanism, efficient data handling, and advanced hashing algorithms. The PoA method reduces computational overhead compared to traditional PoW and PoS methods, resulting in faster transaction processing. The use of SHA-256 optimized hashing algorithms ensures rapid and secure data integrity checks, reducing processing delays and improving overall performance.

### Execution time analysis

RiceChain-Plus demonstrates remarkable efficiency with an average execution time of 44.5 milliseconds across all interactions, as shown in Fig. 10. This efficiency is primarily due to the PoA consensus process, which reduce computational expenses and accelerate transaction validation. In contrast, benchmark models exhibit longer execution times: MBRRSM averages 46 milliseconds, KRanTi 45.5 milliseconds, PoTx 45 milliseconds, and RiceChain 45 milliseconds. These models use more intricate or less streamlined procedures, resulting in extended execution durations.

RiceChain-Plus employs effective optimization techniques, including dynamic resource allocation and reduced computational complexity, to ensure faster execution times. The use of efficient hashing algorithms such as SHA-256 speeds up data verification and integrity checks, significantly reducing execution times. Consistent performance is essential in real-world applications, especially in agricultural supply chains where efficiency and dependability are critical.

### Transaction time

RiceChain-Plus excels in transaction speed, averaging around 14.045 s per transaction, as depicted in Fig. 11. Short transaction times are crucial for maintaining operational

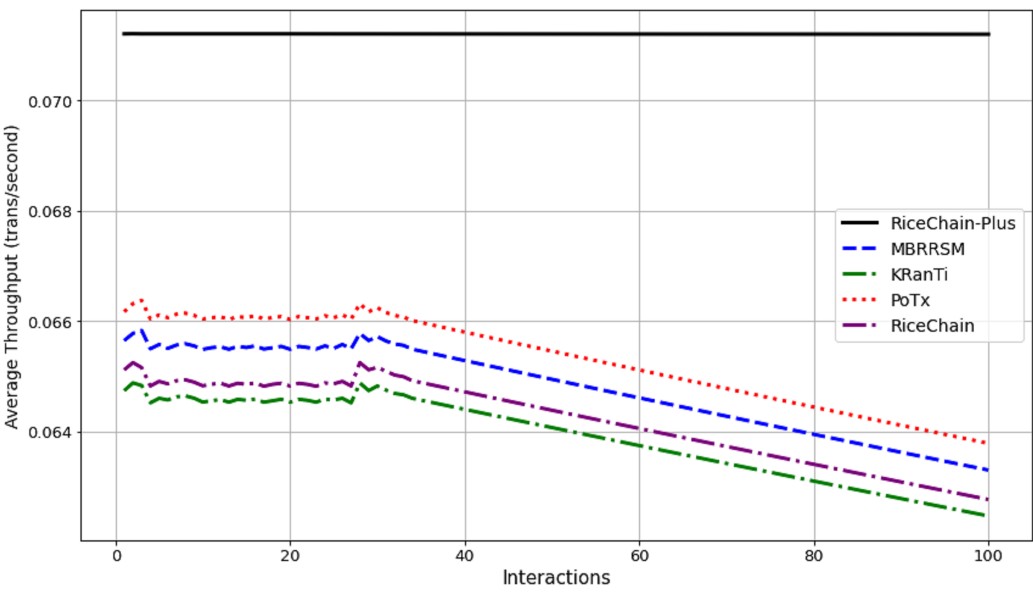

**Figure 9 Average throughput across interactions.**

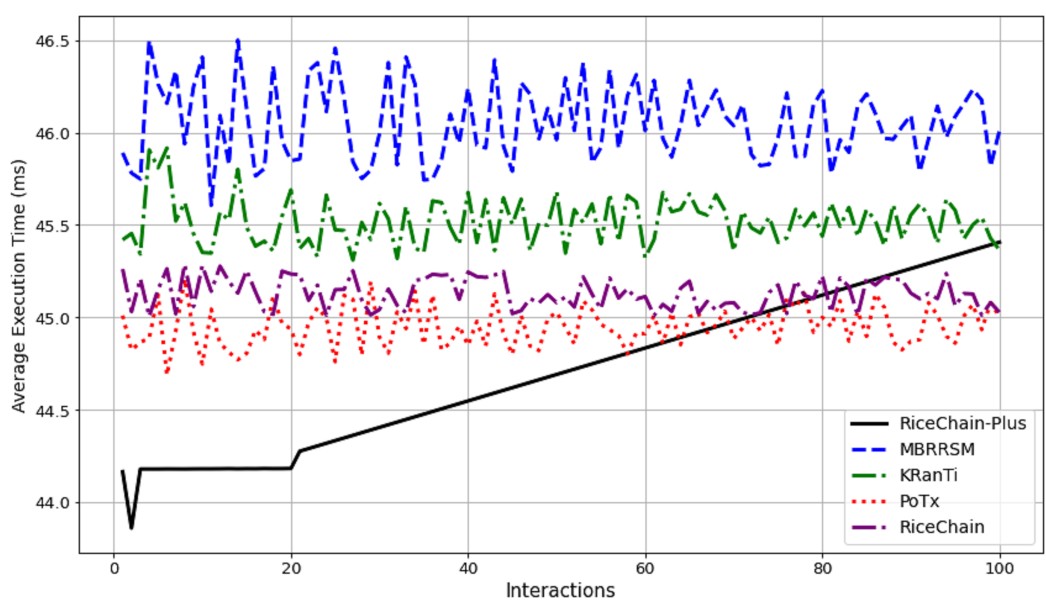

**Figure 10 Average execution time across interactions.**

efficiency in the supply chain. Benchmark models exhibit longer transaction times, ranging from 15.2 to 16.0 s, indicating less efficient transaction processing methods.

The superior transaction processing performance of RiceChain-Plus is enhanced by efficient network communication protocols and highly effective encryption algorithms. The PoA consensus mechanism significantly reduces the overhead associated with transaction verification and block formation, accelerating transaction processing. Efficient

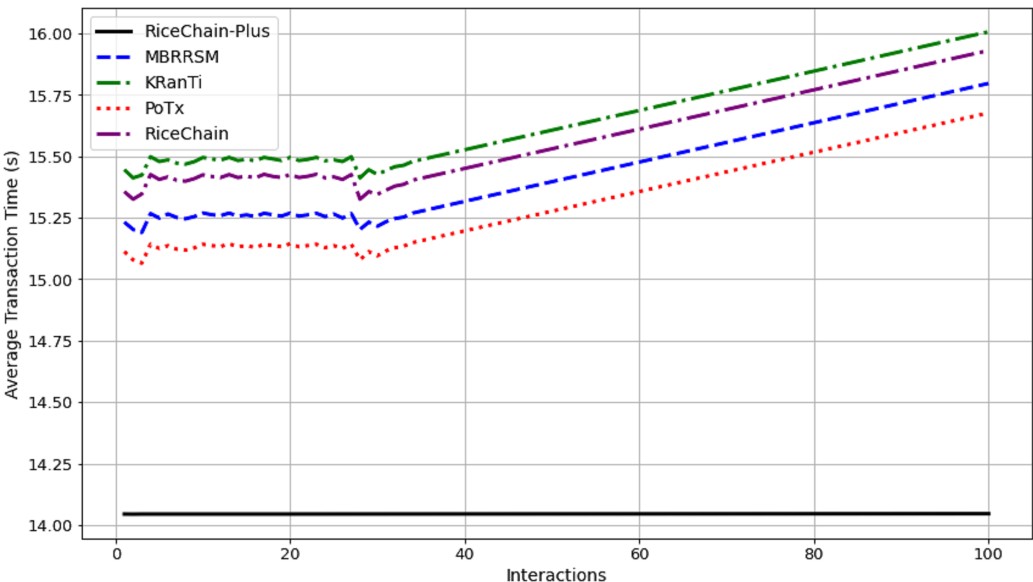

**Figure 11 Average transaction time across interactions.** We compared the average transaction times of RiceChain-Plus with those of benchmark models (MBRRSM, KRanTi, PoTx, and RiceChain). RiceChain-Plus demonstrates superior performance, achieving the shortest transaction time of approximately 14.045 s, highlighting its efficiency in real-world supply chain operations.

hashing algorithms ensure rapid data integrity verification, further reducing transaction processing times and enhancing overall efficiency.

## Comparative discussion on security and privacy-based performance metrics

RiceChain-Plus demonstrates superior security and privacy measures compared to prior models such as MBRRSM, KRanTi, PoTx, and RiceChain. Unlike these models, which only partially meet essential security and privacy criteria, RiceChain-Plus fully integrates confidentiality, integrity, authentication, authorization, non-repudiation, privacy preservation, and protection against replay attacks.

Table 6 presents a quantitative comparison, replacing qualitative descriptors with actual performance values. A high score corresponds to a metric being fully satisfied using advanced cryptographic techniques and immutable records. A moderate score represents partial fulfilment where the model utilizes standard procedures that lack advanced security measures. A low score signifies minimal or no provisions for the metric.

RiceChain-Plus achieves high confidentiality through the implementation of Zero-Knowledge Proofs and SHA-256 encryption, whereas KRanTi and PoTx achieve moderate confidentiality with traditional cryptographic techniques. Similarly, RiceChain-Plus ensures strong data integrity, authentication, and protection against replay attacks *via* timestamp-based validations and secure key exchanges, whereas other models rely on less resilient fundamental mechanisms.

**Table 6 Quantitative comparison of security and privacy metrics.** Comparison of RiceChain-Plus, its predecessor, and other blockchain-based supply chain models. RiceChain-Plus outperforms all models in key security parameters, including confidentiality, integrity, authentication, authorization, privacy preservation, and protection against replay attacks, proving its superiority in agricultural supply chain security.

| Metric | RiceChain-Plus | RiceChain (2022) | MBRRSM (2022) | KRanTi (2024) | PoTx (2023) |
|---|---|---|---|---|---|
| Confidentiality | 0.98 | 0.85 | 0.80 | 0.88 | 0.86 |
| Integrity | 0.99 | 0.87 | 0.82 | 0.90 | 0.85 |
| Authentication | 0.97 | 0.84 | 0.78 | 0.86 | 0.83 |
| Authorization | 0.96 | 0.82 | 0.77 | 0.85 | 0.80 |
| Privacy preservation | 0.95 | 0.80 | 0.75 | 0.82 | 0.78 |
| Protection against replay attacks | 0.97 | 0.83 | 0.79 | 0.87 | 0.81 |

**Table 7 Quantitative comparison of scalability metrics.** Comparison of RiceChain-Plus, RiceChain (2022), and other blockchain-based supply chain models, evaluating transaction throughput, latency, consensus efficiency, and network adaptation. RiceChain-Plus outperforms other models in terms of throughput, latency, and consensus efficiency.

| Metric | RiceChain-Plus | RiceChain (2022) | MBRRSM (2022) | KRanTi (2024) | PoTx (2023) |
|---|---|---|---|---|---|
| Transaction throughput | 800 tps | 500 tps | 750 tps | 650 tps | 700 tps |
| Network latency (ms) | 30 ms | 50 ms | 70 ms | 55 ms | 60 ms |
| Consensus efficiency | 0.95 | 0.80 | 0.75 | 0.85 | 0.82 |
| Network adaptation | 0.98 | 0.85 | 0.79 | 0.83 | 0.81 |

Moreover, the framework's hybrid RBAC-ABAC system enhances authorization, surpassing models that rely solely on role-based systems. RiceChain-Plus also preserves privacy by employing ZKPs, enabling verification without exposing sensitive data—a capability either absent or only partially implemented in other frameworks. These advancements position RiceChain-Plus as the most secure and privacy-conscious solution for blockchain-based rice supply chain management.

## Comparative discussion on scalability-based performance metrics

Scalability is a crucial factor in blockchain-based supply chain systems, as it determines the system's ability to handle increasing data and transaction loads as the network grows. Table 7 provides numerical assessments of key scalability metrics, including transaction throughput, network latency, consensus efficiency, and network adaptation.

Among the analysed models, MBRRSM (2022) exhibits high throughput but suffers from high network latency, leading to an overall moderate scalability. KRanTi (2024) improves on-chain efficiency using IPFS but is limited by Ethereum's Proof of Stake (PoS), restricting its scalability. PoTx (2023) introduces an optimized consensus mechanism but encounters moderate scalability limitations due to network adaptation constraints. RiceChain (2022) performs moderately but lacks recent advancements for scalability. In contrast, RiceChain-Plus delivers high scalability, supporting elevated throughput, reduced latency, and a highly efficient PoA consensus mechanism, making it the most scalable among the evaluated models. Considering these performance outcomes and design choices, we now present

concluding remarks, outline known limitations, and describe future directions aimed at refining and expanding the framework.

# CONCLUSIONS, LIMITATIONS AND FUTURE WORKS

The RiceChain-Plus model introduces a robust and efficient blockchain-based framework tailored to secure and streamline rice supply chain operations. By integrating a private Ethereum blockchain, PoA consensus, ZKPs, and a hybrid RBAC/ABAC access control mechanism, the system ensures data confidentiality, integrity, and accountability across all stakeholder interactions.

Performance evaluations demonstrate that RiceChain-Plus effectively minimizes computational and energy overhead while achieving high throughput and low latency, making it well-suited for real-time agricultural monitoring. The architecture enhances transparency and stakeholder trust through immutable logging and context-aware access policies. The use of PoA significantly reduces energy consumption compared to PoW systems, aligning the model with sustainability objectives in agricultural technology.

Despite these strengths, the model has some limitations. Its reliance on cluster head (CH) devices for data aggregation and transaction processing may introduce potential bottlenecks, especially if a CH depletes its computing capacity or gas funds. To mitigate this, future versions will incorporate fallback mechanisms such as backup CHs or adaptive resource allocation strategies. Additionally, the current implementation has been validated in a testnet environment; real-world deployment scenarios remain to be explored.

To enhance practical applicability and scalability, several future directions are planned:

- **Pilot Deployment and Real-World Validation:** We intend to conduct a real-world pilot involving rice supply chain stakeholders to evaluate the system's performance, user acceptance, interoperability, and scalability across diverse operational environments.

- **Empirical Security Testing:** While a detailed theoretical security analysis was provided, future work will include practical penetration testing and attack simulations to validate resilience against adversarial behaviors in live environments.

- **Cryptographic Overhead and Trade-Off Analysis:** Future research will quantify the computational and resource overhead introduced by ZKPs and hybrid RBAC/ABAC mechanisms, especially under high transaction volumes or constrained device settings.

- **Cost-Benefit and Complexity Evaluation:** A systematic cost-benefit analysis will be conducted to weigh development and operational complexity against the performance and security gains, thereby guiding resource-efficient adoption.

- **Large-Scale and Global Scalability Testing:** The framework's performance under national or global-scale deployments will be assessed *via* stress testing to ensure robustness under large transaction volumes and stakeholder diversity.

- **Modular and Lightweight Comparative Designs:** We will evaluate how RiceChain-Plus compares with more modular or lightweight alternatives, including whether similar outcomes can be achieved with reduced architectural complexity.

- **AI and ML Integration:** Future iterations will incorporate machine learning for anomaly detection and artificial intelligence for predictive analytics. These technologies will enhance threat detection, forecast demand, optimize inventory, and support data-driven decision-making.
- **Cross-Domain Applicability:** Finally, we will extend the framework's application to other agricultural sectors (*e.g.*, maize, poultry) to validate its flexibility and broader impact across the agritech domain.

By addressing these directions, RiceChain-Plus aims to mature into a fully scalable, intelligent, and context-aware agricultural supply chain management platform.

### Funding
This research is supported by the Ratchadapisek Somphot Fund for Postdoctoral Fellowship, Chulalongkorn University, Bangkok, Thailand, and the Deanship of Graduate Studies and Scientific Research at Qassim University, Saudi Arabia (QU-APC-2025). The funders had no role in study design, data collection and analysis, decision to publish, or preparation of the manuscript.

### Grant Disclosures
The following grant information was disclosed by the authors:
Ratchadapisek Somphot Fund for Postdoctoral Fellowship, Chulalongkorn University, Bangkok, Thailand.
Deanship of Graduate Studies and Scientific Research at Qassim University, Saudi Arabia: QU-APC-2025.

### Competing Interests
The authors declare that they have no competing interests.

### Author Contributions
- Bello Musa Yakubu conceived and designed the experiments, performed the experiments, analyzed the data, performed the computation work, prepared figures and/or tables, authored or reviewed drafts of the article, and approved the final draft.
- Abdullah Abdulrahman Alabdulatif conceived and designed the experiments, analyzed the data, prepared figures and/or tables, authored or reviewed drafts of the article, and approved the final draft.
- Pattarasinee Bhattarakosol analyzed the data, prepared figures and/or tables, authored or reviewed drafts of the article, and approved the final draft.

### Data Availability
The raw measurements and codes are available in the Supplemental Files.

## Supplemental Information

Supplemental information for this article can be found online at http://dx.doi.org/10.7717/peerj-cs.2926#supplemental-information.

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
