# Peer review of "RiceChain-Plus: an enhanced framework for blockchain-based rice supply chain systems-ensuring security, privacy, and efficiency"

_PeerJ Computer Science, doi:10.7717/peerj-cs.2926_

## Round 0.1 · original submission · Major Revisions

There are some problems with the correctness and security analysis of the proposed model. It is unclear how the provided results can be evaluated to understand the performance behaviors of the proposed model. The mutual authentication phase needs to be proved by providing a correctness analysis. A notation table should be added to explain the symbols used. The security analysis section needs to be reconsidered. Is there any security analysis model to examine the security-related properties of the proposed model? The security analysis section needs to be rewritten by examining the theoretical aspects. Experimental results and discussion sections should also be detailed by showing the provided results. It also requires a comparison table in terms of performance with related literature proposals. I am not sure Figures 7-11 are important in real-world scenarios. Please add much more information about how the provided results, given in Figures 7-10, should be evaluated. If any source codes are used to obtain the provided results, please add accessible links. Tables 2-3 do not provide meaningful comparisons in terms of examined security metrics. What is high and moderate? In the proposed idea, related primitives, such as hybrid Role-Based Access Contr, zero-knowledge proof, etc., are not well-defined and well-explained to clarify the main contribution of the proposed approach to the literature. Figure 1 needs to be detailed by adding a step-by-step explanation and used primitives. All phases of the proposed model should also be explained in detail.

Reviewer 1 ·

Basic reporting

• Add a brief description of the methodology or technology implemented (e.g., the type of blockchain used and specific cryptographic techniques) to further clarify the study’s scope in abstract.
• Consider adding a brief comparative mention of RiceChain-Plus’s advantages over other blockchain applications in the food and agricultural sector in introduction to contextualize its uniqueness
• Clarify the exact scope of the study’s focus within the rice supply chain, as the introduction is less expressive.

Experimental design

• Improve clarity by elaborating on the role of the fog nodes and detailing their contribution to efficiency beyond data aggregation.
• Consider adding a flowchart or table to represent different adversary types and their specific countermeasures.
• To clarify the security protocol, consider a flowchart illustrating the two-level process or discussing potential real-world limitations, such as device compatibility

Validity of the findings

• Include a baseline comparison with existing blockchain models in similar supply chains to strengthen the performance .
• Mention any future plans for testing in real-world.

Reviewer 2 ·

Basic reporting

a. Clarity: The paper is written with professional English and provides a detailed context regarding blockchain's role in the rice supply chain.
b. Background and Literature: A solid review of related works contextualizes this study within existing blockchain and agricultural supply chain research.
c. Figures and Tables: Diagrams like Figure 1 (showing the proposed framework) and the tables summarizing performance data contribute significantly to the paper’s clarity. However, ensure all figures are adequately labelled and easily interpretable.

Experimental design

a. Research Question: The study clearly states its aim to improve supply chain security, privacy, and efficiency through blockchain technology, addressing a current knowledge gap.
b. Methods: The methods are well-described, allowing for replication. The multi-phase security protocol, including mutual authentication and two-level access control (RBAC and ABAC), is a notable improvement. However, adding more justification for the chosen protocols could strengthen the rationale.

Validity of the findings

a. Data and Analysis: The results appear statistically sound, with robust data on execution costs, energy consumption, and throughput. A comparative analysis with existing models (e.g., MBRRSM and KRanTi) highlights the improved efficiency of RiceChain-Plus.
b. Conclusion: The paper’s conclusions are well-aligned with the findings. However, discussing potential limitations (such as the hypothetical scenario where a cluster head device runs out of funds) would add a realistic perspective to the study’s claims.

Additional comments

7. General Comments:
o The paper’s focus on implementing a private Ethereum blockchain, mutual authentication, and zero-knowledge proofs demonstrates a well-rounded approach to securing the rice supply chain.
o Addressing challenges like scalability and energy consumption is particularly commendable given blockchain’s typical energy demands.
o Consider expanding the discussion on environmental impact or alternative low-energy consensus methods to enhance the paper’s appeal for readers interested in sustainable technology solutions.

·

Basic reporting

1.Introduction Section (Location: Line 22-30): In the section concerning the clarity of research objectives, it is recommended to explicitly highlight the main goals within the introduction. The current description appears somewhat dispersed, so consider adding a clear objective statement or reorganizing this section to make the logic more coherent.
2.Literature Review Section (Location: Line 101-185): In the literature review, although the applications of blockchain in agricultural supply chains are comprehensively introduced, the categorization of the literature does not clearly connect to the proposed framework. It is suggested to restructure the literature review by categorizing it into specific technological challenges (such as privacy, scalability, energy consumption) and explicitly linking these challenges to the contributions of RiceChain-Plus.
3.In the "Adversary model" section of the article, external adversaries and internal adversaries are proposed. It is recommended to add information about which part of the process solves what problem. Or, after the question of an attack is raised, in which part of the following is it resolved.
4.In the "Integrity and message authenticity" section of the article proposes how messages can be stored and transmitted without tampering. How to ensure the credibility of the collected data before it is uploaded to the blockchain? A supplementary explanation is recommended.

Experimental design

1.The steps in the mutual authentication phase after line 355 are numerous. It is recommended to merge the steps executed by each role in sequence.

Validity of the findings

1.Conclusion Section (Location: Line 950-970): In the conclusion, add a stronger emphasis on the broader implications and potential scalability of the framework. Consider including practical benefits, such as increased supply chain transparency and stakeholder trust, to leave a lasting impression on readers.

Additional comments

no comment

---

## Round 0.2 · Major Revisions

I think the contributions of the paper need to be improved. I am not sure using some structure or primitive together means a enough contribution to literature. The methodological contributions must be defined and detailed.  The literature review needs to extension about how proposed solution provides novelty about blockchain technology in agricultural supply chains. You have to highlight and explain how you can ensure the efficiency of prosed framework. There are some weaknesses in Security analysis. "The irreversible characteristic of the hash function guarantees that these values cannot be rebuilt from captured data. Consequently, the adversary is unable to extract significant information or generate authentic messages, hence guaranteeing the protocol's resilience against side-channel attacks  aimed at sensor nodes."  This claim is re-evaluated with the one-way power of the selected hash function. There are some claims like this one. They have to be reconsidered.

You need to define how it differs from RiceChain clearly. You should do this in a separate subsection.

As in the RiceChain example, you should also add the "Gas cost of Ethereum functions in USD and Vulnerability analysis" table for RiceChain Plus.

You should detail separate hands-on performance evaluations and component-by-component comparisons for RiceChain and RiceCainPlus.

Based on Figure 6, explain the transaction flows algorithmically in the proposed framework.

As ZKP, Role-Based Access Control, and Attribute-Based Access Control please explain which algorithms you use in practice and why. If available, add the accessible codes used to obtain the application. Otherwise, please provide details on how you obtained the results.

Give the discussion section under a subheading and present the evaluation of the proposed solution based on the technical, methodological and performance results of the existing solutions.

Tables 4-5 do not give a meaningful comparison. High, moderate and low metrics should be handled by considering numerical values. How did you evaluate execution costs, energy consumption, throughput, execution time, and processing time in Figures 8-11?  How did you obtain the results for the MBRRSM (2022), KRanTi (2024), PoTx 901 (2023) studies?  The entire performance evaluation process should be considered in detail for all metrics.

Reviewer 2 ·

Basic reporting

• Clarity and Language: The paper is written in clear and professional English, with technical terminology that is well explained. Minor grammatical and sentence clarity improvements could further enhance readability, particularly in the abstract and introduction (e.g., avoiding overly long sentences).
• Literature Context: The introduction and literature review provide a detailed context for the research, clearly identifying gaps that RiceChain-Plus aims to address.
• Figures and Tables: Figures are relevant, well-labeled, and clear. Tables (e.g., Table 1 summarizing related works) effectively compare previous studies. Ensure all symbols in Tables 2 and 3 are consistently defined in the text.
• Raw Data: Raw data from the simulations is not fully visible in the PDF, but performance metrics are presented systematically. Ensure underlying datasets are included as per PeerJ guidelines.

Experimental design

• Scope and Originality: The paper proposes an original framework tailored to blockchain-based rice supply chains, addressing critical limitations (e.g., scalability, privacy, energy efficiency). This fits well within the journal’s scope.
• Research Question: The research question is well defined: the development of RiceChain-Plus to improve security, privacy, and efficiency in rice supply chains. The authors clearly state the knowledge gap in existing blockchain systems.
• Methods: The methodology, particularly the integration of Zero-Knowledge Proofs (ZKP), PoA consensus, and hybrid RBAC/ABAC models, is described in detail. However:
o Some steps in the security and mutual authentication phases are overly technical without clear summaries for broader audiences.
o Include a simplified flow diagram summarizing the experimental setup to improve accessibility.
• Replication: The methods provide sufficient detail to replicate the experiments, especially the performance evaluations using Ethereum Virtual Machine (EVM).

Validity of the findings

• Data Robustness: The study includes robust performance evaluations (execution cost, energy consumption, throughput, and execution time) compared to four benchmark systems (MBRRSM, KRanTi, PoTx, RiceChain).
• Statistical Soundness: While performance metrics are provided, no statistical analysis or significance tests are discussed. Include details of the variability across simulations (e.g., standard deviations or error margins).
• Conclusions: The conclusions are well supported by the results. RiceChain-Plus demonstrates efficiency, scalability, and security improvements over prior models.

Additional comments

Strengths:
1. Comprehensive Framework: The integration of ZKP, PoA, fog nodes, and hybrid RBAC/ABAC enhances security, scalability, and energy efficiency.
2. Rigorous Evaluation: Performance comparisons against well-known benchmarks add credibility to the results.
3. Real-World Relevance: The focus on rice supply chains addresses practical challenges, providing valuable contributions to agricultural blockchain applications.
Weaknesses and Suggestions:
1. Clarity:
o Simplify technical discussions, particularly in the mutual authentication and cryptographic phases.
o Summarize results and methods visually (e.g., concise tables, simplified flowcharts).
2. Statistical Analysis:
o Include variability or confidence intervals in performance metrics to strengthen claims.
o Discuss the impact of sample size (e.g., 100 transactions over 100 minutes) on results.
3. Future Work:
o The authors briefly mention integrating machine learning for anomaly detection. Elaborate on how this could further enhance the system.

Reviewer 4 ·

Basic reporting

In this paper, the authors propose RiceChain-Plus, a blockchain-based framework aimed at addressing security, privacy, and efficiency issues in the rice supply chain. The system integrates the Proof of Authority (PoA) consensus mechanism, a private Ethereum blockchain, and advanced cryptographic techniques to ensure scalability, security, and energy efficiency. The proposed framework has been evaluated through security analyses and experimental assessments, measuring transaction time, energy consumption, and data privacy.

• The language and expression of the paper should be revised to align with academic standards, ensuring greater clarity and readability. Specifically, the Introduction and Literature Review sections contain overly indirect phrasing and long sentences, which should be refined.
• There are inconsistencies in the writing structure, requiring a thorough revision.
• The Introduction does not sufficiently emphasize the significance of the paper. It should explicitly outline the shortcomings of existing solutions and clearly explain how the proposed system addresses these gaps.

Experimental design

• The role of nodes in data collection and validation is well explained theoretically. However, additional technical details are needed regarding their practical feasibility (e.g., data collection frequency, latency, hardware requirements).
• Some technical aspects remain ambiguous. For instance, while the paper states that each Cluster Head (CH) generates its own secret key, it does not clarify how this key is cryptographically generated.
• Further elaboration is required, particularly regarding the system’s practical implementation:
o There is no technical explanation regarding whether secret keys are stored securely in a hardware security module or a trusted storage environment.
o The paper mentions the use of Secure Multi-Party Computation (SMPC) but does not specify the exact algorithms employed.
o Multi-Factor Authentication (MFA) is referenced, yet the specific authentication methods supported (e.g., SMS, OTP, biometric verification) are not detailed.
o The paper states that Role-Based Access Control (RBAC) and Attribute-Based Access Control (ABAC) models are utilized. However, it does not clarify how authorization policies are updated, who is responsible for these modifications, or how these processes are audited.
o While the paper claims that the system is secure and mentions cryptographic techniques, it lacks a comprehensive analysis of how the model was tested against potential attacks. The system's resilience to real-world malicious threats remains uncertain.

Validity of the findings

• The paper indicates that each scenario involves 100 transaction interactions over 100 minutes among stakeholders. However, it does not justify why these tests are considered sufficient. Additional experiments with larger datasets should be conducted.
• Limited information is provided regarding the real-world applicability of the proposed model. The paper should elaborate on how the simulations align with real-world usage scenarios.
• The paper asserts that RiceChain-Plus excels in terms of security and scalability. However, the classification levels (High, Moderate, Low) lack clearly defined criteria. The methodology used to determine these classifications should be explicitly stated.

Additional comments

I recommend major revision.

Reviewer 5 ·

Basic reporting

✔ Professional and clear language used throughout
✔ Strong literature support and contextual background provided
✔ Well-structured article with high-quality figures and tables
✔ Self-contained with relevant results aligned to hypotheses
✔ Includes formal results with well-defined terms and mathematical proofs

Recommendation
The manuscript meets all criteria for basic reporting and is well-prepared for publication. Minor refinements in grammar and readability could further enhance clarity, but no major revisions are necessary.

Experimental design

Original primary research within the journal’s scope
✔ Well-defined research question addressing a real-world problem
✔ High technical and ethical standards in cybersecurity and blockchain application
✔ Detailed and replicable methodology, allowing future studies to build on this work

The only minor improvements could be:
• Clarifying parameter choices in the experiments (e.g., Why were 100 transactions over 100 minutes chosen for testing?)
• More discussion on real-world deployment beyond the simulated environment.
Overall, the study is technically rigorous, methodologically sound, and highly relevant, making a valuable contribution to blockchain-based supply chain security.

Validity of the findings

✔ Meaningful replication is encouraged, with a well-stated rationale and contribution to literature.
✔ All underlying data are robust, statistically sound, and controlled, ensuring reliability.
✔ Conclusions are well-stated, aligned with research questions, and strictly based on supporting results.


Additional comments

✔ Overall, the paper is well-prepared for publication and provides a significant contribution to blockchain-based agricultural supply chains.
✔ Only minor revisions are needed to improve statistical validation, real-world deployment discussions, and grammar clarity.
✔ Addressing the above points will enhance the paper’s impact, readability, and practical significance.

---

## Round 0.3 · Minor Revisions

Please finalize your paper by considering the required reviewers' comments.

Reviewer 1 ·

Basic reporting

The revision i have told to authors, it is totaly satisfied and i accept the manuscript

Experimental design

The experimnetal esign is well explained with all the tables and figures

Validity of the findings

The findings of the study is good at this level of blockchain implementation.

Additional comments

none

Reviewer 2 ·

Basic reporting

Strengths:
Addresses a real-world problem in agricultural supply chains.

Strong technical architecture with multiple layers of security and privacy.

Use of PoA and private Ethereum is appropriate for controlled environments.

Performance results are clearly reported and favorable.

Includes security analysis to support claims of resilience.

Experimental design

The framework was likely tested in a simulated environment or local testnet (e.g., Ethereum test network). Benchmarked against existing models in terms of computational cost, speed, and energy efficiency.
To strengthen the credibility and allow others to build upon the work, providing: Source code or smart contract repository. Detailed benchmarking setup (e.g., blockchain node specs, transaction types, network conditions) would greatly enhance reproducibility.

Validity of the findings

The claim is strong, but would benefit from more transparency about the types of attacks tested.
The paper highlights improved performance and security but doesn’t discuss potential trade-offs such as: Overhead introduced by ZKPs and hybrid access control.

Complexity in implementation and maintenance.

A discussion on the cost-benefit balance (in terms of development effort, scalability vs. complexity) would improve the practical understanding of the system's deployability.

Additional comments

Comment: While RiceChain-Plus introduces a novel integration of several security and privacy techniques, many of these components (PoA, ZKP, RBAC/ABAC) are not new individually. The innovation lies more in how these tools are combined and applied to the rice supply chain than in the invention of new techniques. he system is claimed to be scalable, but there’s limited insight into how it would perform at a national or global supply chain scale.

Suggestion: Future work could explore how this integration compares with modular or alternate designs, or whether the same outcomes could be achieved with fewer or more lightweight components. Including stress tests or simulations under heavy load would support scalability claims and provide more robust validation.

Reviewer 5 ·

Basic reporting

The paper is well-structured

Experimental design

The research question is well-defined, and the study addresses a relevant problem.

Validity of the findings

The results are logically presented and supported by data. Some could be strengthened with additional statistical analysis

Additional comments

Ensure smooth transitions between sections for better readability

---

## Round 0.4 · accepted · Accept

According to the reviewer's comments, your paper is now ready for publication. Please ensure that the final version has no formal errors.

Reviewer 2 ·

Basic reporting

ok

Experimental design

ok

Validity of the findings

ok

Additional comments

ok

Reviewer 4 ·

Basic reporting

The authors have generally made a careful and thorough revision based on the feedback. In particular, they explained the technical parts more clearly, gave more detail about the authorization system, and removed confusing terms like SMPC and MFA. These changes have made the paper easier to understand and technically stronger.

Experimental design

-

Validity of the findings

-